

# Birth and closure of the Kallipetra Basin: Late Cretaceous reworking of the Jurassic Pelagonian – Axios-Vardar contact (Northern Greece)

Lydia R. Bailey[1,2], Vincenzo Picotti[2], Maria Giuditta Fellin[2], Filippo L. Schenker[3], Miriam Cobianchi[4], Thierry Adatte[5]

[1]Department of Geosciences, University of Arizona, Tucson, AZ 85721, USA
[2]Department of Earth Sciences, Institute of Geology, ETH Zurich, 8092 Zürich, Switzerland
[3]Institute of Earth Sciences, University of Applied Sciences and Arts of Southern Switzerland, 6952 Canobbio, Switzerland
[4]Department of Earth and Environmental Sciences, University of Pavia, Pavia, 27100, Italy
[5]Institute of Earth Sciences, University of Lausanne, 1015 Lausanne, Switzerland

*Correspondence to*: Lydia R. Bailey (lydiabailey@email.arizona.edu)

**Abstract.** Some 20 Ma after the Late Jurassic to Early Cretaceous obduction at the eastern margin of Adria, the eroded Pelagonia (Adria) – Axios-Vardar (Oceanic Complex) contact collapsed, forming the Kallipetra Basin, described around the Aliakmon river near Veroia (Northern Greece). Clastic and carbonate marine sediments deposited from early Cenomanian to end Turonian, with abundant olistoliths and slope failures at the base due to active normal faults. The middle part of the series

is characterized by red and green pelagic limestones, with minimal contribution of terrigenous debris. Rudist mounds in the upper part of the basin started forming on the southwestern slope, and their growth was competing with a flux of ophiolitic debris, documenting the new fault scarps affecting the Vardar Oceanic Complex (VOC). Eventually, the basin was closed by overthrusting of the VOC towards the northeast and was buried and heated up to ~ 180 ˚C. A strong reverse geothermal gradient is recorded by illite crystallinity and zircon fission tracks, with temperatures increasing up-section to near 300 ˚C at the tectonic

contact with the VOC. We interpret this anomaly as due to fluid migration from deeper sources and/or shearing affecting the porous and permeable deposits during early burial diagenesis. This study documents the reworking of the Pelagonian – Axios-Vardar contact, with Cenomanian extension and basin widening followed by Turonian compression and basin inversion. Thrusting occurred earlier than previously reported in the literature for the eastern Adria, and shows a vergence toward the northeast, at odds with the regional southwest vergence of the whole margin.

## 1. Introduction

The Hellenides are an integral segment of the main Alpine-Himalayan orogenic belt (Fig. 1). They have recorded polyphase Alpine deformation since the Middle Jurassic, when they were involved in the obduction of imbricate oceanic units over the eastern Apulian margin (Bernoulli and Laubscher, 1972; Zimmerman and Ross, 1976; Schmid et al., 2020). In the internal Hellenides, continuous convergence led to collision of continental promontories with Eurasia in the Late Jurassic-Early

Cretaceous that built a metamorphic crustal-scale orogenic wedge involving the Pelagonian zone and Rhodope (Burg et al., 1996; Ricou et al., 1986; Schenker et al. 2014). In the Late Cretaceous, the metamorphic thrust sheets of the Pelagonian zone



were exhumed to shallow depths. This is testified by a cooling below ca. 240 °C from 83 Ma onwards along the northern margin of the Pelagonian zone (Most, 2003; Schenker, 2013), from 54 Ma to the south (Lipps et al., 1998, 1999; Coutand et al., 2014), and by the deposition of metamorphic Pelagonian detritus in a Late Cretaceous basin (Kossmat, 1924; Schenker et

al., 2015). During the Late Cretaceous-Eocene, thrusting resumed in the internal Hellenides (Godfriaux et al., 1988; Schermer, 1993) and progressively migrated SSE to the external Hellenides (e.g. Aubouin, 1973). Finally, in the Oligocene-Miocene, a pulse of extensional exhumation occurred (Schermer et al., 1990; Lacassin et al., 2007; Coutand et al., 2014; Schenker et al., 2014).

In the Pelagonian zone and adjacent units, the record of this orogenic system in the time interval between collision in the Early

Cretaceous and resumed thrusting in the Late Cretaceous-Early Cenozoic remains sparse and discontinuous leading to many and sometime contrasting large-scale interpretations (Fig. 2). To elucidate part of these controversies, this study investigates a small Upper Cretaceous basin that formed on both ophiolitic and continental units along the eastern Pelagonian margin (Fig.1) and was overthrusted by serpentines of a Jurassic oceanic floor (the ophiolitic fragments now laying west of the Pelagonia zone named Axios/Vardar/Almopias zone by Schenker et al., 2015). The basin collected coarse detritus bearing Pelagonian

metamorphic and ophiolitic rocks and deposition was followed by deformation of the sediments, by thermal conditions that locally partially or totally reset the cooling ages, and by cooling during the Late Cretaceous (Schenker et al., 2015; Sharp and Robertson, 2006). However, the stratigraphic evolution and the depositional age of this basin are so far only partially constrained. Moreover, it remains unclear how thermal conditions (temperatures > ca. 240 °C) and deformation in this basin relate to the apparent tectonic quiescence associated with extensive Late Cretaceous cooling recorded elsewhere in the

Pelagonian zone (Schenker et al., 2015) and to the diachronous and complex tectonic evolution of the Hellenides. Finally, the timing of opening and sealing of the basin and the tectonic environment of deposition are fundamental to unravel the Late Cretaceous interval of the long history of accretion, subduction, arc-magmatism and large-scale extension in the Hellenic subduction system (*e.g.,* Burg, 2012; Jolivet and Brun, 2010; Ring et al., 2010 and references therein).

This study uses conventional geological mapping techniques, stratigraphic analysis, illite crystallinity, and low temperature

thermochronology to obtain new constraints on the tectonic evolution of the eastern margin of the Pelagonian zone and to unravel the Late Cretaceous detrital record. Our data indicate that the Upper Cretaceous basin was shallow and tectonically active as testified by the presence of olistoliths, large gravitational features such as rotational growth faults and slumping, and early diagenetic deformation. Rudist bioherms were accumulated on the shallow slopes of the basin with flank deposits dipping into the basin. The bioherms were terminated through environmental restriction or burial due to increased serpentinite sediment

input from the eroding ophiolitic complex to the south-southwest. Moreover, based on illite and petrographic data, we find an inverted, high, non-linear geothermal gradient related to a heating event, which likely occurred during the early Late Cretaceous.





## 2 Background

### 2.1 Large-scale tectonic setting

Following the Variscan Orogeny and Permian strike-slip and extension (Schenker et al., 2018), the Permian-Triassic rifting led to the creation of the Tethys and its seaways, namely the Pindos, Vardar/Maliac and Meliata basins, that continued to open during the Triassic to Early Jurassic (Bernoulli and Laubscher, 1972; Papanikolaou, 2009; Schmid et al., 2008). The convergent motion between Eurasia and Adria led to a northward intra-oceanic subduction in the Vardar in the Early-Middle Jurassic that saw the production of magmatic arcs to the north (Bortolotti et al., 1996; Burg, 2012; Dimitrijevic, 1982). In the Late Jurassic, there was south-westward obduction of the Tethys ophiolite (Axios/Vardar/Almopias zone) onto the passive continental margin of the Pelagonian zone to the south (Bernoulli and Laubscher, 1972; Dimo-Lahitte et al., 2001). Jurassic-to-Lower Cretaceous sediments were imbricated during the accretion of the ophiolitic units (Bortolotti et al., 2005; Robertson and Dixon, 1984). Continuous crustal shortening caused the accretion of Rhodope by the latest Jurassic-Early Cretaceous and of the Pelagonian zone by the Early Cretaceous (Figs. 1 and 2; Burg et al., 1996; Moulas et al., 2017; Ricou et al., 1998; Schenker et al., 2014). The buried Pelagonian basement experienced regional amphibolitic-facies metamorphism to the north (U-Pb zircon metamorphic ages of migmatites at 130-117 Ma; Schenker et al., 2015, 2018) and an upper greenschist- to blueschist-facies metamorphism to the south (Ar-Ar ages on muscovite at 100-85 Ma; Lips et al., 1998; Schermer et al., 1990). The non-metamorphosed Pelagonian sediments to the south show a sedimentary Aptian-Albian hiatus (~120-100 Ma) over lower Aptian deformed flysch and bauxitic laterites testifying deposition and deformation in the frontal part of the wedge, followed by growing topography during the accretion of the lower crustal units of the Pelagonian zone (Nirta et al., 2015; 2018).

Thereafter, transgressive Cenomanian-to-lower Campanian limestones and deep-water Paleocene turbidites unconformably overlay the eroded Pelagonian and Axios/Vardar/Almopias imbricated units (Papanikolaou, 2009) attesting to deepening below sea-level of the Rhodope-Pelagonian crustal-scale orogenic wedge. Moreover, during the Late Cretaceous-to-Eocene and locally since the Campanian, the imbrication of the Axios/Vardar/Almopias units resumed at several locations in relation to thrusting with vergence mostly to the SW. This has been documented in the central-eastern Vardar (Paikon Window; Godfriaux and Ricou, 1991; Bonneau et al., 1994; Brown and Robertson, 2003; Katrivanos et al., 2013), in the northwestern Vardar (Grubić et al., 2009; Ustaszewski et al., 2009), in the northeastern Pelagonian zone (Kilias et al., 2010), in the southern Pelagonian zone (Baumgartner, 1985) and in the Pindos zone (e.g. Aubouin, 1959, 1973; Papanikolaou, 1997). Continuous convergence up to the Neogene progressively deformed the continental margin of the Adriatic plate into southwest-verging fold and thrust sheets (Fig. 1; Channell and Hovarth, 1976). Final exhumation of the stacked crustal and oceanic piles occurred through extensional metamorphic domes between the Eocene in the north and late Neogene in the south (Burg, 2012; Gautier et al., 1993, 1999; Jolivet and Brun, 2010; Lister et al., 1984).



## 2.2 Main geologic features of the eastern Pelagonian margin

The Pelagonian basement consists of deformed: (i) orthogneisses crosscut by leucogneiss and leucogranites; (ii) mafic amphibolite bodies; and (iii) interlayered marbles (Schenker, 2013). Cooling of the Pelagonian core complex carapace rocks may have started at or after collisional doming at $118 \pm 4$ Ma (U-Pb metamorphic zircon ages, Schenker et al. 2018). $^{40}Ar/^{39}Ar$ white mica ages from the Pelagonian gneisses show a younging toward the dome core from 111-100 to 80-64 Ma that witness the slow exhumation and cooling of the deeper units of the basement (Schenker, 2013).

The Axios/Vardar/Almopias unit includes a mélange zone made of tectonically superimposed marbles, serpentinites (ophicalcites), flysch-phyllitic series, volcanoclastic sediments, amphibolites and carbonatic sequences imbricated during the Late Jurassic obduction over the Pelagonian zone to the west (e.g. Ferriere et al., 2016; Ricou and Godfriaux, 1995; Sharp and Robertson, 2006; Smith et al., 1975). In the study area, the Axios/Vardar/Almopias unit is represented by serpentinites that are referred to as the Vardar Ophiolitic Complex (VOC), which consists of at least 5 lithologies: (i) ophicalcites; (ii)

hydraulically brecciated serpentinite; (iii) sedimentary serpentinite breccia; (iv) sedimentary serpentinite breccia with platform carbonates; and (v) foliated serpentinite and limestone. Ferromanganese chert nodules within the VOC, dated further to the south at approximately 175 Ma by Chiari et al. (2013), attest the involvement of this Jurassic oceanic floor in the intra-oceanic Tethys subduction and subsequent obduction.

   On the eastern margin of the Pelagonian zone, relatively thick packages of Upper Cretaceous carbonatic and siliciclastic

sediments with both a Pelagonian and ophiolitic provenance unconformably cover the VOC and the Pelagonian basement (Papanikolaou, 2009; Schenker, 2013; Schenker et al., 2015; Sharp and Robertson, 2006). The sediments belong to a sedimentary basin that here is referred to as the Kallipetra Basin (Fig. 3). It formed as an elongate NNW-SSE oriented belt overlying the VOC and the Pelagonian continent. In this basin, the presence of reworked Lower Cretaceous *Orbitolinids*, *Globotruncana sp.* and mid Turonian *Helvetoglobotruncana helvetica* indicates deposition during the Cretaceous (Schenker

et al., 2015). Based on these depositional ages, a ZFT age of 67 Ma from an orthogneiss boulder (sample 10-128) at the top of the Kallipetra Basin was previously interpreted as indicating a very short lag time between cooling of the dome and deposition in the basin. Two more samples (10-029 and 10-130) from the Kallipetra basin were interpreted as possibly partially to non-annealed.

   From the early Oligocene, an overall southwestward tectonic denudation from shallow depths is documented in the Kallipetra

basin by AFT ages of 32.7 Ma (sample 10-128) and in the Pelagonian basement by ZFT ages of 24 – 20.7 Ma and AFT ages between 22.9 and 16.1 Ma (Schenker et al., 2015).



## 3 Methods

### 3.1 Geologic mapping and stratigraphy

Geological mapping and structural analysis were conducted to reconstruct the geometry of the basin and the ductile and brittle
deformation that affected the Kallipetra Basin and the VOC. The paleogeography, depositional environment, and age of the
sedimentary sequences were determined based on stratigraphic logging, optical microscopy and biostratigraphy. Planktonic
foraminifera and nannoplankton were used to establish ages of the sedimentary package. Simple smear slides were produced
using standard techniques to retain the nannofossil assemblages and original sediment composition. Quantitative analyses were
carried out using a polarizing light microscope at a magnification of 1250x.

### 3.2 Illite crystallinity

The Kübler Index of illite crystallinity is a method used to determine grade in metapelitic sequences by measuring the changes
in shape of the first dioctahedral illite-muscovite basal reflection at a 10-Å X-ray diffraction (XRD) spacing (Kübler and
Jaboyedoff, 2000). To analyze illite crystallinity, bulk-rock mineralogy was obtained through the conventional powder XRD
method using the ARL Thermo X'tra powder diffractometer at the University of Lausanne. Samples were then de-carbonated,
followed by the extraction of <2 µm clay fraction and 2-16 µm fraction that were used for further analysis. Oriented samples
were prepared by sedimentation on a glass slide from the suspended fraction. Samples were first air-dried (AD), and then
treated with ethylene glycol (EG) to recognize any overlapping effect of smectite peaks. XRD diffractograms were performed
on both the AD and EG treatments. The full width at half-maximum height (FWHM) of the illite 10-Å XRD peak that is
measured on both AD and EG clay samples (<2µm size fraction) gives the Kübler Index (KI) (Kübler and Jaboyedoff, 2000).
KI is expressed as $\Delta°2\theta$ CuK$\alpha$. The air-dried KI value is used for the determination of metapelitic zones and approximate
temperatures. It should be noted that the KI does not serve as a precise geothermometer, but provides a qualitative indicator of
stages that phyllosilicates may have reached through metastable mineral reactions (Abad, 2007; Merriman and Peacor, 1998).
Significant asymmetrical peak broadening, caused by a tail in the 10-Å peak and produced by the presence of smectite and
expandable mixed layers, is reduced following EG treatment (Abad, 2007). These peaks may indicate the presence of detrital
illite, which gradually decreases with burial and essentially disappears in the anchizone (Kübler and Jaboyedoff, 2000). The
decrease of KI values with increasing metamorphic conditions and temperatures is a consequence of the increase in the number
of layers and disappearance of expanding layers.  The Neuchâtel IC scale was calibrated with the Lausanne diffractometer and
therefore produced anchizone limits of 0.18° and 0.36°$\Delta°2\theta$ CuK$\alpha$, which we use in this study (Jaboyedoff et al., 2000).

### 3.3 Zircon fission track dating

Two of five collected samples provided enough zircons to date using zircon fission track (ZFT) analysis: V1503 and V1504.
These samples integrate our previous samples (10-128, 10-129, 10-130; Schenker et al., 2015). The new samples were collected
with the aim of revealing the full age distribution, which in our previous samples was limited by the low number of available



zircons. To this goal, the new samples were > 5 kg each. All the samples consist of arenites, conglomerates, and sandstones. Zircons were separated from the whole rock by initial SELFRAG fragmentation, followed by density-based liquid separation

using a Wilfley water table and heavy-liquid separation. The heavy fraction was passed through the Frantz magnetic separator stepwise to remove magnetic minerals from the zircons. Zircons were embedded in PFA Teflon and the prepared mounts were polished to expose the smooth internal zircon surfaces. The polished mounts were etched using a eutectic mix of NaOH and KOH to preferentially damage the fission tracks, enabling them to be fully revealed for optical analysis. To reveal the whole age distribution, we prepared up to four mounts per sample that we etched at very short time steps of 3.5 hours. Fully etched

zircons were first recognized after 10.5 hours and then we etched the remining mounts to 14 hours and 17.5 hours.

## 4 Results

### 4.1 The Kallipetra Formation: facies and boundaries

The study area is divided into 3 units: (1) the Pelagonian basement; (2) a stratigraphic unit that we name the Kallipetra Formation, described here for the first time; and (3) the VOC. The Kallipetra Formation is the focus of this study and consists

of several lithofacies that collectively characterize a sedimentary basin (Fig. 3). Most field data were collected along two composite stratigraphic sections (the Kallipetra and Sfikia sections, Fig. 4).

The base of the basin is exposed close to the contact with the Pelagonian basement. Locally, the latter consists of a thick package of white, foliated cataclasite (Fig. 3). Directly overlying the cataclasite is a very dark hydraulically brecciated serpentinite, shortly followed by pebbly sandstones and well bedded dark grey limestones. Elsewhere, the base of the basin is

characterized by a thick package of serpentinite-rich conglomerates, breccias, and minor amounts of dark grey limestone (Fig. 4a). The conglomerate is clast-supported and poorly sorted, with a dominance of sub-rounded to rounded clasts greater than 15 cm. The conglomerate is composed of dark green to black serpentinite (~90%), dark grey limestone, marble, and orthogneiss clasts and a fine-grained serpentinite matrix. Thickly bedded, poorly sorted calc-arenites stratigraphically overlie the serpentinite conglomerate. These are openly folded on the meter scale and bedding is deformed around large olistoliths of dark

grey veined marble and serpentinite. The occurrence of olistoliths decreases significantly up section (Fig. 4). In the southeast, the basal contact is sharp and consists mostly of marls, shales and subordinate calc-arenites of the same kind as those observed in the central part of the basin, which are described below (Fig. 4b).

Calc-arenites are observed throughout the basin, typically at intermediate stratigraphic levels (Fig. 4b). The arenites range from fine- to coarse-grained, are medium to thickly bedded, and often display slumping folds. Locally, these folds and the

synsedimentary gravity faults show a top-to-the NE vergence. The quartz content varies with location, with the highest proportion of quartz being in the north-west region of the study area. Locally, the calc-arenites consist of medium- to coarse-grained poorly sorted pebbly sandstones with 1-6 cm sized clasts of red arenite and red-pink carbonate. Very distinctive thinly bedded and laminated red and green marly limestones occur at intermediate-to-high stratigraphic levels (Fig. 4b). The red layers range from 2-5 cm thick, and green layers typically range from 0.5-2 cm thick.


Towards the top of the basin, massively bedded conglomerates and breccias are often interbedded with the calc-arenites and pebbly sandstones, and consist of limestone, bioclastic limestone, arenite, marl, serpentinite, mudstone, and calcareous mudstone as rounded to sub-rounded clasts in a calcareous matrix.

Lateral variations in facies occur frequently, the most evident being the changes observed from the north-western to the central and south-eastern portions of the mapping area. In the north-west (Fig. 4b), the stratigraphy is dominated by coarse to pebbly

sandstones, breccias, and conglomerates whereas shaley-limestones, marls, and mudstones prevail in the south-east (Fig. 4a). In the northwest, lithic fragments of quartz, gneiss, and marble are major components of the coarse sediments, with quartz content ranging from 45% at the base to 90% up section (310 m, fig. 4b), where serpentinite forms a minor component. In addition, olistoliths and evidence of slumping are frequent at high stratigraphic levels in the northwestern sector (Fig. 4b). This differs greatly from the southeastern sector (Fig. 4a), whereby slumped calc-arenites with olistoliths appear only at the base of

the section, and the average quartz content is lower.

The top of the Kallipetra basin is marked by the occurrence of rudist mounds, five of which, some hundreds of meters thick, were identified in the study area. The mounds produce prominent cliffs and dome-like structures in the topography. Each mound can be separated into 4 different facies associations (Fig. 5): (i) the serpentinite and Kallipetra carbonate breccia (SKB); (ii) the mound core; (iii) mound flank; and (iv) the mound top.

The SKB is a sub-angular, moderately sorted, clast supported breccia that is poorly bedded and massive. Clasts comprise of serpentinite, dark grey limestone, rudist-rich microsparite, pink micrite, and minor lithic fragments like quartz, feldspar, and some dark pyroxenes. The rudist microsparite and pink micrite clasts are identical to the mound core. The matrix is composed of a fine- to medium-grained calcareous arenite. *Orbitolinids* were discovered in a clast by Schenker (2013). The SKB is usually found on the southern side or lithostratigraphically below the mound.

The mound core is characterized by light grey to pink, massively bedded micrite and microsparite, in which float abundant whole rudists. *Hippurites* and *Radiolitid* rudists are present along with encrusting sponges and echinoderm fragments. Rudists are scattered throughout the mound and seem to have no preferred orientation. Vertical calcite veins and en échelon veins are frequently observed at the margins of the mound core.

The mound flank is a heterogeneous lithology that varies with distance from the mound core and location within the basin. In

general, a moderately sorted, clast-supported breccia containing large, angular clasts of rudists, red limestone, greenish marls, micrite, and minor serpentinite clasts occurs closest to and on the northern side of the mound core. The number of clasts decreases into a matrix-supported breccia with a marly, green-colored matrix. The serpentinite content gradually increases up-section, and gravel-sandstones contain >60% serpentinite in addition to red microsparite clasts and rudists from the mound core. A sharp sub-vertical boundary often separates the mound flank and the mound core. The mound flank facies differ slightly

throughout the area depending on the location of the mound core. The flank of the northernmost and youngest mound is first characterized by a massive clast-supported breccia consisting of micrite, rudists, sponges, and echinoderm fragments, dissected by neptunian dykes, with onlapping red pelagic marls. Differential compaction structures can be observed in the pelagic sediments where a stratigraphically higher mound core overlies them. Secondly, the clast-supported breccia passes rapidly into



a ~34 m thick sequence of marly limestones only seen on top of the northernmost mound. The proportion of marls within the

mound flank gradually increases from the southerly mounds to the northernmost mound.

Stacking of serpentinite-rich breccias always occurs on the southern slope of the rudist mounds. Flank deposits, either marls or a succession of sandstones and breccias, dip away from the mound core always on the northern mound side.

The mound top, where observed fully, is approximately 6 m thick and is stratigraphically overlying the mound core (Fig. 5). It generally consists of several meters of very poorly sorted, angular to sub-angular gravel of serpentinite and quartz within a

white calcareous matrix. A thin layer of rudist-rich, elongated carbonate clasts overlies the gravel. There is a gradual transition into a clast-supported conglomerate with a reddish calcareous matrix, plus arenite and minor serpentinite clasts. The clasts of this conglomerate are very deformed, where the VOC tectonically overlies them. The full stratigraphy of the mound top was only observed at one, the southernmost, rudist mound (Fig. 5).

### 4.2 Biostratigraphic data

Although significant amounts of sample were collected for biostratigraphic analysis, nearly all of them were barren, or included dissolved, silicified, or recrystallized nannoplankton and foraminifera making most species indistinguishable. Table 1 summarizes the recognizable planktonic foraminifera that were only found near the northernmost mound (Asomata Quarry).

The Orbitolina found in sample M2-TS3, *Mesorbitolina pervia* (A. Arnaud, personal communication)*, has an older stratigraphic distribution than most of the ages displayed in Table 1, with the minimum age being in the basal late Aptian.

However, the fossils are deformed and not perfectly preserved, suggesting that they have been reworked and are supplied by the VOC (Fig. S1). Indeed, Schenker (2013) discovered Lower Cretaceous Orbitolina in the VOC, located very close to the tectonic contact with the Kallipetra Basin. Therefore, this sample is excluded from discussions about the depositional age of the Kallipetra Basin.

Species abundance and totals of calcareous nannofossil were semi-quantitatively evaluated as F = frequent and R = rare. In the

studied section (M2), the major calcareous nannofossil events in stratigraphic order are as follows: the first occurrence of *Quadrum gartneri, Eprolithus octopetalus,* and *Eprolithus eptapetalus* (sample M2/2, 50 cm from the bottom of the section); the first occurrence of *Eiffellithus eximius*, and first and last occurrence of *Kamptnerius magnificus* (sample M2/6B, 2 m from the bottom of the section) (Fig. S2)*.

*Q. gartneri, E. octopetalus,* and *E. eptapetalus* can be correlated with the UC7 zone in the Turonian stage, giving the base of

the M2 section a minimum age of 93.6 Ma (Burnett et al., 1998). *E. eximius* and *K. magnificus* can be correlated with the base of the UC8 zone in the Turonian stage.

### 4.3 Post-sedimentary structural data

In the marls and marly limestones of the Kallipetra Basin, the foliation is mostly parallel to the bedding and defined by flat and elongated quartz clasts and clay minerals. Bedding and foliation dip at a very low angle either to the NW or to the SE due

to bending around an axis plunging shallowly to the NE (Fig. 6a, b; Table S3). Stretching lineations are observed mostly on



lamination surfaces in fine-grained marls, limestones and mudstones and they are formed by the alignment of elongated clay minerals. Mineral lineations occur mainly in the Pelagonian basement where the long axis of elongated quartz and feldspar crystals are aligned. In all the lithologies the lineations strike NNE-SSW at low dip angles (< 20 °; Fig. 6b). Stretching lineations along with asymmetric interlayered boudinaged beds indicates a top-to-the NNE shear sense within the basin (Fig

7a).

The top of the Kallipetra basin is tectonically covered by the VOC. In the profiles A-A' and B-B' (subparallel to the NE-SW lineations, Fig. 8), the contact appears preferentially flat and dips with shallow angle (< 15°) to NE. Along the profile C-C' (orthogonal to the lineations, Fig. 8), the contact is bent over the mound cores and flanks, forming an undulate surface. To the NE, the Kallipetra sediments overlie the VOC forming tectonic duplexes (Profile B-B', Fig. 8). The shear zone in the footwall

of the VOC is characterized by 2 to 6 m thick foliated cataclasites and by a strain gradient visible through the increase in the intensity of the foliation. The cataclasite is usually white in colour, clay rich, and often features floating carbonate and/or serpentinite blocks in the matrix. Conglomerates below the contact between the VOC and rudist mounds show a 3 m-thick strain gradient from almost non-deformed clasts at the bottom, to cigar shaped and highly elongated clasts at the contact (Fig. 7c; Profile B-B', Fig. 8). The orientation of the long axis of the cigars is sub-parallel to the stretching lineations observed

throughout the study area suggesting that the deformation during the tectonic emplacement of the VOC was penetrative within the basin. Shear bands, stepover structures and sigma-clasts in the cataclasite indicate a top-to-the NE tectonic movement (Fig. 7b) synthetic to the intra-basin shearing. However, new growth of syntectonic chlorite along the main contact shows that shearing below the VOC occurred at higher thermal conditions with respect to the intra-basin deformation.

Late normal faults trending W-E to NW-SE and two transtensional to strike-slip faults crosscut the Pelagonian basement, the Kallipetra Basin and the VOC (Fig. 3). Most steep normal faults plunge to the NE (Fig. 8). Low-angle fault zones are observed within the VOC dipping approximately 35° towards the NNE with a normal top-to-NE shear sense. This later extensional phase may have locally reactivated the major tectonic contact. The dextral strike-slip component of the fault along the valley of the Aliakmon River reached approximately 50 m in the south and just a few meters in the north. Another transtensional fault

causes a small normal displacement of approximately 50 m that uplifts the northernmost mound core.

### 4.4 Illite crystallinity data

37 samples for illite crystallinity analysis were taken up-section in the north part of the Kallipetra Basin (Kallipetra section). Four samples were unsuitable for illite crystallinity analysis as the <2μm portions contained no illite. The remaining samples have KI ranging from 0.091 to 0.388 (Fig. 4b; Table 2).

Stratigraphically higher samples have KI ranging from 0.09 to 0.25, and stratigraphically lower samples have KI of 0.38. The KI appears to increase down-section for samples containing only non-detrital illite. The sample with the lowest KI of 0.091 is characterized by an XRD pattern that reveals the presence of chlorite. The sample with KI of 0.141 contains paragonite which indicates epizone conditions.





The samples containing detrital illite are limited to stratigraphic heights between 300 and 350 meters and show a large range
of KI between 0.14 to 0.383. Non-detrital illite, on the contrary, is mostly confined to stratigraphic heights above 400m. Using
the anchizone limits as calibrated for our lab (see section 3.2; Jaboyedoff et al., 2000), and given the fact that the effects of
detrital micas disappear in the anchizone (~200-300 °C), the results indicate that the Kallipetra sediments experienced higher
temperatures (lower KI values) closest to the tectonic contact with the VOC. This is supported by the presence of paragonite
at the top of the section. The KI values subsequently increase away from the contact, indicative of an inverse geothermal
gradient from >300 °C to 100-200 °C within ~165 m (Fig. 4b).

### 4.5 Zircon fission track

We collected our samples along a down-section direction within the Kallipetra Basin: the only ones that produced enough
countable zircons are from close to the contact with the VOC. Results are reported in table 3. The two successful samples are
from the same location but from two different layers: a sandstone and a conglomerate. Both rocks are sheared and contain
newly formed chlorite (Fig. 9). In sample V1504, 61 grains could be counted on the 10.5 and 17.5 hour etch. In sample V1503,
79 grains could be counted on four mounts with the 3 different etch times (10.5, 14 and 17.5 hours). Both samples consist of
multiple age populations as attested by the $\chi^2$ test that gives a probability of 0 % (Galbraith, 1981). V1504 has grain ages in
the range from 75 to 660 Ma with a central age of 156 +/- 10 Ma and V1503 from 74 to 468 Ma with a central age of 177 +/-
13 (Fig. 10; Table S4). At least two to three age populations can be identified using the software DensityPlotter (Vermeesch,
2012). The age distribution of sample V1504 has two major peaks: one centered at 150 +/- 6 Ma contains 84% of the grains,
the other at 433 +/- 68 Ma is formed by 16% of the grains. The main younger peak might represent the sum of two populations
at 128 Ma +/- 11 Ma and at 183 +/- 19 Ma, respectively.  The age distribution of sample V1503 has a major peak with a
pronounced shoulder and a long tail towards older ages. The central peak represents the largest age population that consists of
70% of the grain and that has an age of 158 +/- 14 Ma. The shoulder represents a minor population centered at 105 +/- 14 Ma
with 19% of the grains. A third population might be located along the tail of the distribution at 252 +/- 50 Ma.

## 5 Discussion

### 5.1 Onset and evolution of the Kallipetra Basin

The orthogneiss- and serpentinite-rich composition of the previously described basal cataclasite suggests that it was formed
prior to or at the same time with the formation of the Kallipetra Basin and mainly at the expense of the Pelagonian basement
and of the VOC. These normal faults crosscut duplicates of Pelagonian mylonitic marble, and must be younger than ca. 120
Ma (Schenker, 2014).  Normal faulting during or following the exhumation and doming of the Pelagonian zone from the late
Early Cretaceous (Schenker, 2014; Schenker et al., 2015) probably contributed to subside the deformed wedge below sea level
to create the basin. The extension produced an uneven paleogeography in which one slope of the basin was prevalently built
on Pelagonian basement, and the other on the VOC, allowing detritus from both lithologies to enter the depression.





Serpentinite-rich conglomerates represent the first sediments deposited within the Kallipetra Basin through subaerial erosion of the VOC, which created an uneven topography. The occurrence of conglomerates followed by a succession of calc-arenites at the base of the basin indicate shallow marine depths. Marble olistoliths and slumping at the base of the Sfikia section indicate instability during the first phases of basin formation and the presence of a proximal steep slope in which gravitational instability drove slumping. In the north-western part of our study area, the presence of orthogneiss blocks, the dominance of quartz,

feldspar, gneiss, and marble lithics in the sediments, and the lack of such components in the southeast, suggest an intrabasinal high, emergent land, or continent existed northwest of the Kallipetra Basin, where the Pelagonian basement was exposed. In the south-eastern part of the study area, the dominance of silty limestones, marls, lime mudstones, and the rare presence of olistoliths indicate that there is a deepening of the basin away from the north-west. The mid part of the Kallipetra Formation is devoid of serpentinite coarse detritus, suggesting the initial fault scarps were smoothed by sediments. This expansion of the

basin toward the southeastern slope formed by the VOC produced the transgression recorded in the study zone.

## 5.2 The rudist mounds: facies and evolution at the slope of the Kallipetra Basin

Rudists constituted more than 60% of reef frames during the Aptian and Albian and became the most dominant frame-building organism in the Late Cretaceous (Scott, 1988; Voigt et al., 1999). Widespread tectonic extension combined with eustatic continental flooding occurring around the Cretaceous Tethyan Ocean allowed the growth of broad carbonate platform

complexes, on slopes from a few degrees up to 40° (Gili et al., 1995). Previously described carbonate mud mounds and rudist biostromes have some similarities to the rudist mounds observed in the study area (e.g. Camoin, 1995; Negra et al., 1995; Sanders, 1998; Sanders and Höfling, 2000; Sanders and Pons, 1999). It has been suggested that bioerosion processes leading to pervasive micritization of invertebrates may result from endolithic microorganism activity, accounting for part of the lime muds (Camoin, 1995). Alternatively, Camoin (1995) also suggest the lime muds are formed through in situ precipitation

promoted and induced by microbial activity, and/or the decay of microbial communities. The latter is the most likely option regarding these mounds, with the dense micrite deposited as leiolite (sensu Riding, 2000) in a microbe-rich upper slope. Microbial mud mounds were common in the Late Cretaceous of the western Tethys, especially on the shelf/ramp rimming the Adria microplate (e.g. Picotti et al., 2019). The distinct dome shape of the mounds built mainly from lime mud, suggests that the mounds themselves were sites of increased carbonate productivity. Furthermore, the upward growth of rudists is said to be

an adaptation to environments with positive net sedimentation rates (Gili et al., 1995), which may be the case for the Kallipetra Basin. The sub-vertical and sharp nature of some of the contacts between the mound cores and the flank deposits, and the presence of breccia bodies, suggest early diagenetic consolidation allowing stability of the steep mound slopes. By combining the observations of breccia bodies stacking up against the southern flanks of mounds, and the presence of stratigraphically underlying slumping and mass-flow deposits, the rudist mounds were built on a slope environment. Open shelf or shallow-

water platforms, conditions suggested by Scott (1988) and Camoin (1995), are unlikely for the Kallipetra Basin due to its dynamic and tectonically active history. The termination of the rudist mounds, found in the upper part of the stratigraphy of





the Kallipetra Basin, occurred by environmental restriction due to gradually increased sediment input from the approaching ophiolite talus (Sanders & Pons, 1999; Sanders & Höfling, 2000).

The mound flanks consist of a succession of sandstones, breccias, and occasionally marls. Our observation of polymictic
breccias on the south-southwestern mound slopes bear an important paleogeographic meaning. In this case, the presence of serpentinite clasts indicates the breccias were not formed solely from erosion and collapse of mound flanks, but rather they were derived from an ophiolitic source up slope from the mounds, possibly associated to new fault scarps in the south-southwestern slope of the Kallipetra Basin. On the northern flanks of the mounds, the sediments display a shallower dip and are interfingered with the mound talus breccias. The youngest and northernmost mound at Asomata displays at the northeastern
flank pelagic marls and limestones, suggesting deeper bottom conditions to the N and NE. On the other hand, the absence of serpentinite detritus in the mound flanks other than the southern ones documents a shadow effect of the mounds with respect to the south-southwestern provenance of the serpentinite clasts. This evidence corroborates the presence of a slope dipping to the north/northeast. The increasing serpentinite content in the sandstones and breccias up-section suggests that the ophiolitic source was moving closer to the mound structure and gradually providing material to the slope. The positioning of the flank
deposits and the northeastward migration of the two or three youngest mounds, with the highest - and therefore youngest - mound being at the Asomata quarry in the northeast of the study area, suggest that in the upper part of the Kallipetra stratigraphy, there was a movement of the ophiolite (VOC) from SSW to NNE providing at first the slope for the growth of the mounds, then the burial for them (Fig. 12).

**5.3 Stratigraphy and age of the Kallipetra Formation**

The Kallipetra Formation was deposited on top of the eroded Pelagonian continent and obducted ophiolite after the end of the collision-related burial and cooling/exhumation of the Pelagonian zone at ~116 Ma (Schenker, 2014).

The red and green limestones can be loosely correlated across the Kallipetra Basin and they first occur at approximately ~250 m from the base of the Kallipetra construction road section, and 400 m from the base of the Sfikia section. These facies indicate deposition in a deep, calm pelagic environment. The distinct red and green alternations are typically attributed to bottom-water
redox cycles (*e.g.* Luciani and Cobianchi, 1999), developed around the Upper Cenomanian OAE2 (Luciani and Cobianchi, 1999; Mort et al., 2007; Negri et al., 2003). This event coincides with the global Cenomanian-Turonian sea level transgression, therefore explaining the relative absence of clastic input in this interval, that could represent the deepest stage of the basin development. Indeed, this should be the timing of the onlap of the Kallipetra basin toward the southern ophiolitic slope.

Rudist mounds and breccias are lacking in the Kallipetra sediments found on top of the VOC (Fig. 3 and 8), whereas fine
sediments dominate. The fine material suggests that the tip of the VOC was under sea level, with a transgressive trend and widening of the basin, allowing onlap of fine material over the VOC slopes at the same time as red and green marl deposition. During this time, the source area for sediments is moving away. This Kallipetra material overlying the VOC is somewhat separated from the main Kallipetra Basin sediments, possibly through a structural high or as perched basins (Fig. 12).



Alternatively, these deposits represent one flank of the basin that was subsequently tectonically emplaced over the basin,
suggesting this movement was just a few kilometers and a local event.

*Helvetoglobotruncana helvetica* indicates that the marls overlying the mound at the top of the Kallipetra stratigraphy are lower
Turonian. This agrees with the other nannoplankton that we found in the overlying section that are lower to middle Turonian.
The youngest proven depositional age of the Kallipetra Basin is ~ 92 Ma, but the top 35 m of hemipelagic marls are barren,
therefore we cannot exclude a latest Turonian or Coniacian age for the very top of the Kallipetra basin.

The stratigraphic thickness between the red and green marls, and the marls adjacent to the mound core containing the
*Helvetoglobotruncana helvetica,* is 200 m. By using an age of 93.9 Ma (Cenomanian – Turonian boundary) for the red and
green marls (Cohen et al., 2013), and the youngest age of *Helvetoglobotruncana helvetica* - 91.3 Ma (BouDagher-Fadel and
Price, 2019)- for the marls at the top of the section, then the average sedimentation rate of the basin infill is 0.08 mm year$^{-1}$, a
value compatible with the recorded mixture of pelagic and terrigenous sediments. Assuming a constant sedimentation rate
during infilling of the basin, the first sediments deposited at the bottom of the basin are lower Cenomanian. Therefore, there
was approximately 20 Ma of erosion and/or subsidence between the final stage of the collisional doming of the Pelagonian
basement and subsidence and deposition of the first Kallipetra sediments. This 20 Ma time interval agrees with the Aptian-
Albian sedimentary hiatus (~120-100 Ma) over a lower Aptian flysch observed further south in the Pelagonia zone, which is
attributed to a growing topography during collision and subsequent subsidence (Nirta et al., 2015; Nirta et al. 2018).

**5.4 Zircon fission-track age distribution and thermal overprint**

Our new ZFT samples come from the top of the Kallipetra Basin where the depositional age should not be older than 92 Ma
and therefore should be younger than the ZFT central ages of our samples, which range between 156 and 177 Ma (Fig. 10).
However, both ZFT samples contain a few young grains with ages overlapping with the depositional age of the Kallipetra
Formation (Fig. 10). They are located a few hundred meters to the south of a previous sample, 10-029, that is adjacent to the
contact with the VOC (Schenker et al., 2015; Fig. 10). The age range of this sample is from 52 to 340 Ma, and it consists of
16 grains that define only one age population centered at 92 +/- 9 Ma, in overlap with the depositional age. Thus, all these
samples can be interpreted as partially to non-annealed, and the sample closest to the contact with the VOC has the youngest
age. They are all from clastic sediments that contain newly formed chlorite. No illite crystallinity data are available as the rock
type of the ZFT samples do not allow the illite method to be applied. However, we observed a NE-SW metamorphic gradient
along the section where we collected the dated samples together with others that unfortunately provided no zircons. This
gradient is indicated by the fact that the sandstones of V1503, V1504 and 10-029 contain newly formed chlorite, whereas the
sandstones (V1505) located 4 km to the SW towards Sfikia show only detrital minerals (Fig. 9). The presence of newly formed
chlorite in samples V1503, V1504 and 10-029 suggest temperature conditions ≥ 250 ºC, which could be within the partial
annealing zone (PAZ) for natural zircons bearing radiation damage (Reiners and Brandon, 2006). However, the temperature
range of the PAZ depends not only on the degree of radiation damage of the zircons but also on the rate of heating and cooling
such that during a short-lived heating event, followed by rapid cooling, higher temperatures are needed to obtain fully reset



ages. The fission-track kinetic parameters in natural zircons are constrained only based on exposed fossil annealing zones (Brandon et al., 1998) such that modeling their time-temperature history would not give any deeper insight on the conditions that could have produced the observed age distribution.

Two more samples were previously dated along the Kallipetra section where we collected our new illite crystallinity data (Fig. 10). There, 20 grains from the sample at the top (10-128) of the section define an age range between 39 and 102 Ma and a central age of 67 +/-4 Ma; 26 grains from the lower sample (10-130) have ages from 40 to 158 Ma and centered at 72 +/- 5 Ma. These samples come from the top of the Kallipetra section and they were previously interpreted as non-annealed. However, based on the revised depositional age of the Kallipetra basin documented here, the central ages of these samples result younger

than the depositional age, but their age ranges partly overlap with the depositional age. Thus, these samples can be defined as partially to fully annealed and they are younger than the samples 10-029, V1503 and V1504. The illite crystallinity data indicate temperatures up to $\geq 300\,°C$ towards the top of the Kallipetra section. The top ZFT sample 10-128 is from a higher stratigraphic location than that of the illite samples, whereas sample 10-130 comes from the same location as the uppermost illite samples. Thus, the ZFT samples along the Kallipetra section should have been subject to $T \geq 300\,°C$ but we cannot say if and how much

higher these temperatures could have been relative to the other samples.

The different ZFT age ranges and central ages hint to highly variable degrees of annealing. Our petrographic and illite crystallinity data constrain a strong, inverse, vertical (up-section) thermal gradient but they cannot discriminate possible lateral gradients across the basin. However, they indicate that the Kallipetra Basin has been subject to temperatures that locally could have totally or partially annealed our samples. Whether these gradients are reflected by the ZFT central ages or grain-age

distributions must be carefully pondered against other factors that could also affect our results. In fact, we processed the new and the old samples purposefully in different ways because, while processing the previous set of samples, we realized that the low number of available zircons limited the applicable etch procedure, which was not optimal to reveal the full age spectra of our samples. However, even though at the time we opted for an etch procedure aimed at maximizing the young grain ages, our results indicated that the annealing degree of our samples might have been incomplete. With the new samples, we aimed at

verifying the degree of annealing by maximizing the zircon yield that allowed applying a multiple etch procedure. This in turn revealed that in fact there are wide age distributions in the new samples, which include non-reset ages, and this confirmed our previous observations on a partial degree of annealing. Unfortunately, our new data do not answer all the questions concerning the ZFT ages in the study area but highlight a complex thermal and annealing record.

**5.5 The inverted geothermal gradient in the Kallipetra Basin**

The KI data constrain an inverse geothermal gradient at the top of the Kallipetra Basin from > 300 °C at the tectonic contact to 100-200 °C ~165 m below the overridden VOC (Fig. 11). Stratigraphically below this zone, the sediments reached only deep diagenetic conditions. The newly formed syn-tectonic chlorites in the top sediments at the base of the VOC further testify high (> 200 °C; Beaufort et al., 2015) and inverse temperatures that peaked at the time of deformation.





The illitization reaction (i.e. the conversion of smectite-rich I-S into illite-rich I-S) is also dependent on the availability of K$^+$

ions, sometimes requiring enhanced K$^+$-rich fluid circulation (Dellisanti et al., 2008). However, the corroboration of temperatures between the chlorite-in reaction and the KI values suggests that the K+ ion were available during the increase of the metamorphic conditions and that other mechanisms influencing the crystallinity of illite such as shear related recrystallization (Árkai et al., 2002; Merriman and Peacor, 1998) were less important in controlling the KI values. Hence, it is likely that the KI values represent metamorphic temperatures recording a ~~dramatic~~ inverse geotherm.

Sedimentary strata within thrust belts are known to sometimes experience transient thermal histories, and 'sawtooth' geotherms with inverse metamorphic fronts from the base of the hanging-wall into the footwall have been recognized in a series of thrust systems (Furlong and Edman, 1989; Graham and England, 1976). The inverted thermal profiles by fault zones require an extra heat source in addition to conductive relaxation after burial (e.g. Barton and England, 1979; Graham and England, 1976). In the Kallipetra Basin several additional heat sources can be envisaged: (i) heat advection through emplacement of the hot VOC

or percolation of hot fluids and (ii) in-situ heat production through shear heating (e.g. Barton and England, 1979; Camacho et al., 2005; Graham and England, 1976; Hooper, 1991; Mase and Smith, 1984). In shear zones these mechanisms act contemporaneously, and one heat source may dominate over the other depending on the rheology of the rocks and on viscosity, strain rate, thickness and dip angle of the shear zone. With fast plate velocities (>2 cm/a), the heat surplus is mainly advected by the thrust sheet when viscosities are $< 10^{-19}$ or is produced by in-situ shear heating when viscosities are $> 10^{-20}$ (Duprat-

Oualid et al., 2015). With < 2 cm/a, the heat is mainly conducted (Duprat-Oualid et al., 2015), hence at low velocities dramatic inverse thermal gradients such the one of the Kallipetra Basin are probably created by inputs of continuous or spasmodic hot fluids. However, the data collected so far does not allow to calculate the convergence rate, strain rates and thickness of the VOC potentially transporting the heat.

The sedimentary history suggests that the closure of the Kallipetra Basin by the VOC occurred just after the deposition of the

ophiolitic debris that buried the mounds, when the sediments were porous, permeable and saturated. Accordingly, the viscosity of the sediments was low, probably reducing the contribution of heat derived by shear heating, unless the velocities were extremely high. At the microscale, the mechanical feedback between deformation and pore fluid pressurization along fault zones may lead frictional heating to generate fast and transient thermal perturbations with rises of temperature of up to > 500 °C (Vredevoogd et al., 2007) but the influence of these short-term pulses on the long-term thermal overprint results difficult

to quantify. Hence, with "normal" convergence rates and low viscosities, the heat surplus is most likely allochthonous either coming from the transported ophiolitic sheet or from the rise of hot fluids. Changes in basin geometry, sediment compaction, uplift, and tectonic loadings from overriding tectonic sheets can all contribute to continuous changes in the groundwater systems, especially in foreland basins (Ge and Garven, 1989). Significant thermal perturbations require focusing of fluids in a spatial or temporal sense, for example, along fault zones (Deming et al., 1992). The overriding unit over the Kallipetra basin

would have allowed fluid focusing and differential loading that caused any fluids to flow in the direction of tectonic transport (Fig. 10).





Overall, our data document an inverse thermal gradient of the Kallipetra Basin, pointing to a heating event that produced a transient, inverse, non-linear and disturbed geotherm. Although the ultimate causes of this heat are not clearly established, the non-reset to partially reset FT ages testify that this syn-tectonic heating event formed in the Late Cretaceous, during the closure

of the basin in the Turonian. Cooling slightly postdates the deformation as the youngest ZFT population is older than the Turonian closure.

### 5.6 Sealing of the Kallipetra Basin and large-scale implications

According to the sedimentary evolution of the basin and to the kinematic indicators along the tectonic contact and within the basin, the direction of tectonic transport of the VOC sealing the Kallipetra Basin in the Turonian was from the SSW to the

NNE (Fig. 12). During the Late Cretaceous-to-Eocene, along the eastern Pelagonian margin, the dominant deformation at the regional scale is SW-verging thrusting (e.g. Schenker et al 2015). Biverant thrusting occurred locally but later in time during the late Late Cretaceous (Brown and Robertson, 2003; Katrivanos et al., 2013). Thus, the sealing of the Kallipetra basin occurred earlier than or in the very early phase of this regional deformation event, although the direction of tectonic transport of the VOC above this basin is opposite to the common SW-vergence of thrusting. This apparent difference may be explained

by a localized basin inversion rather than a regional tectonic event that predated the start of the regional convergence in the Late Cretaceous or Campanian at the earliest (Aubouin, 1973; Baumgartner, 1985; Godfriaux and Ricou, 1991; Bonneau et al., 1994; Papanikolaou, 1997; Brown and Robertson, 2003; Grubić et al., 2009; Ustaszewski et al., 2009; Kilias et al., 2010; Katrivanos et al., 2013; Schmid et al., 2020). In this scenario, the inverted geothermal gradient in the Kallipetra basin was likely produced by heat advection related by the overriding VOC or by hot fluids. The presence of hot fluids could be related

to the not too far occurrence of Late Cretaceous oceanic crust that is documented in the Dinarides (Prelević et al., 2017; Ustaszewski et al., 2009), in the Cyclades (Fu et al., 2012) and in Crete (Langosch et al., 2000). However, these basaltic magmatic centers are all dated to the Campanian and this is likely later than the sealing of the Kallipetra Basin that seemingly took place during the Turonian. Overall, independently from the source of heat that caused the inverted geothermal gradient, the closure of the basin anticipates the beginning of resumed ophiolitic imbrication in this sector of the internal Hellenides in

the Turonian.

### Conclusions

The evolution of the Kallipetra Basin documents the transition from extension to compression during the early Late Cretaceous along the eastern margin of the Pelagonian zone in northern Greece. The history of the Kallipetra basin can be summarized as follows:

-    The sediments of the Kallipetra Basin were deposited between the early Cenomanian (~100 Ma) and the latest
             Turonian (~90 Ma) over the VOC and the Pelagonian basement in a depression deepening to the east and north-east.
             The depression formed initially by extension as testified by normal faults at the base of the basin.



- As the basin widened, a topographic high located to the NW and exposing Pelagonian basement rocks became the main source of siliciclastic detritus to the basin. Carbonate sediments were produced by pelagic organisms and by

rudist mounds growing on the southwestern slopes of the basin (Fig. 12). The basin widened and deepened to the point when no clastic input reached it. This time might correlate with the global Cenomanian-Turonian sea level transgression.

- The terrigenous input was later renewed, and the main source were ophiolitic rocks to the south or south-west, which provided breccias stacking up against the southern flanks of the rudist mounds. The progressive increase of detrital

input restricted the environments of the rudist mounds.

The ophiolitic rocks overrode the Kallipetra basin from the SW, causing uneven deformation of its sediments (Fig. 12). Thrusting occurred along with the circulation of hot fluids close to the tectonic contact and imprinted a high inverted geothermal gradient that caused illitization and partial-to-total annealing of the fission tracks in detrital zircons close to and increasing towards the top of the basin. Deformation, illitization and zircon-fission track annealing occurred during the

Turonian and were followed by cooling in the late Late Cretaceous, anticipating the beginning of the resumed tectonics in this sector of the internal Hellenides by about 10 Ma.

*Data availability.* Additional data is available from the corresponding author upon request.

*Supplement.* The supplement related to this article is available online at: xx

*Author contributions.* LRB is the primary author, conducted the study for her MSc thesis at ETH Zurich and wrote the paper. VP contributed to the interpretation and provided supervision and mentorship in all aspects of this study. MGF conducted the fission track analysis, provided contributions to Sect. 1, 3.3, and 4.5 and to interpretations of data, and also provided mentorship. FLS conceptualized the original research goals and aims of this study, and contributed to the interpretations, introduction and background sections of this paper. MC prepared samples for and conducted all planktonic foraminifera

analysis for this study. TA prepared samples for and conducted illite crystallinity analysis and gave constructive suggestions on the final paper draft.

*Competing interests.* The authors declare that they have no conflict of interest.

*Acknowledgements.* We thank xxx for constructive reviews of this study. A special thanks goes to Tiemen Gordijn for field assistance in the Kallipetra Basin, and to the community of Sfikia for hosting us. We thank A. Arnaud and D. Bernoulli for

help with fossil identification.

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



**Figures and Figure Captions**


**Figure 1: Location of the Hellenides and study area in the Alpine Mediterranean chain. (Modified from Burg, 2012).**






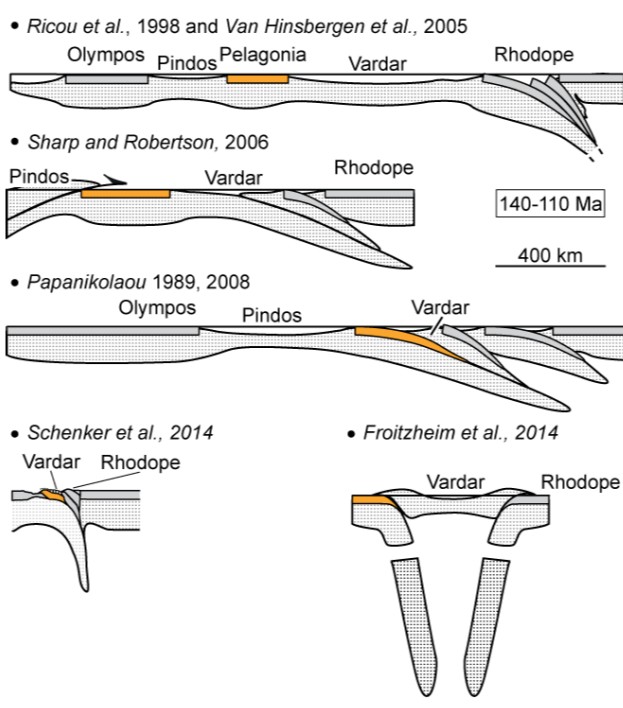


**Figure 2: Geodynamic interpretations of the Hellenides in the early Cretaceous according to different authors. Note that there is no consensus on the early Cretaceous geodynamic framework of the basins between the Pelagonian zone and the Vardar domain.**








**Figure 3: Geological map of the study area. Dark blue circles indicate locations of illite samples, black circles indicate location of ZFT samples and their respective ages.**







**Figure 4: (a) Stratigraphic column taken along the lower road leading to Sfikia, located directly south of the Aliakmon River, and stratigraphic locations and ages of ZFT samples; (b) Stratigraphic column taken along the Kallipetra Monastery construction road, located north of the Aliakmon River, the ages and stratigraphic locations of ZFT samples, and illite crystallinity samples. Illite crystallinity samples are plotted against Kubler Index and diagenetic zone.**


**Figure 5: Stratigraphic sections of the Vardar ophiolitic complex, mound top and mound flank.**






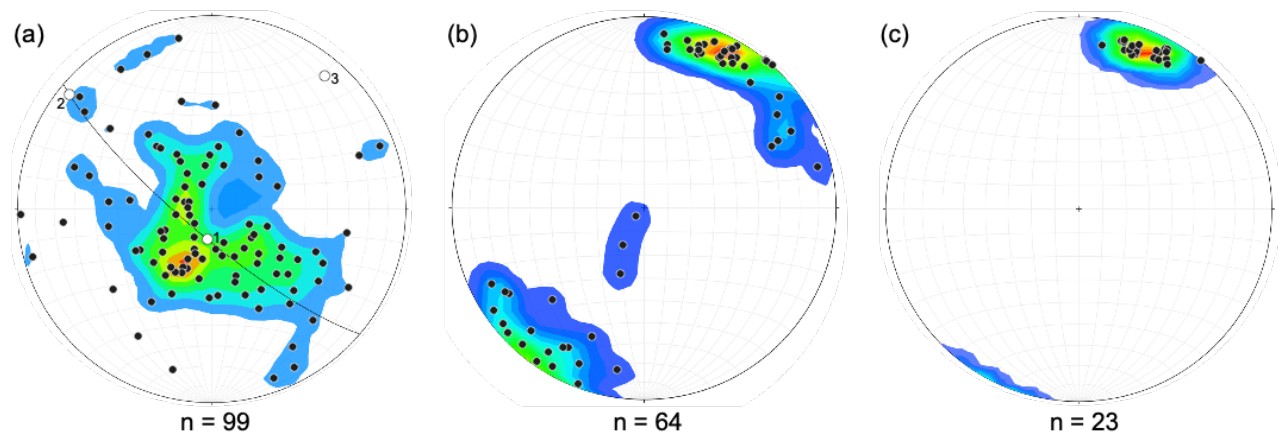

**Figure 6: Lower hemisphere stereoplots of: (a) foliation poles whereby measurements were taken from marls and limestones; (b) mineral and stretching lineation measurements of the mapped area; and (c) stretching lineations of the strained conglomerate at grid reference N40° 27' 23" E022° 15' 00". These measurements do not include foliations observed in foliated cataclasites.**

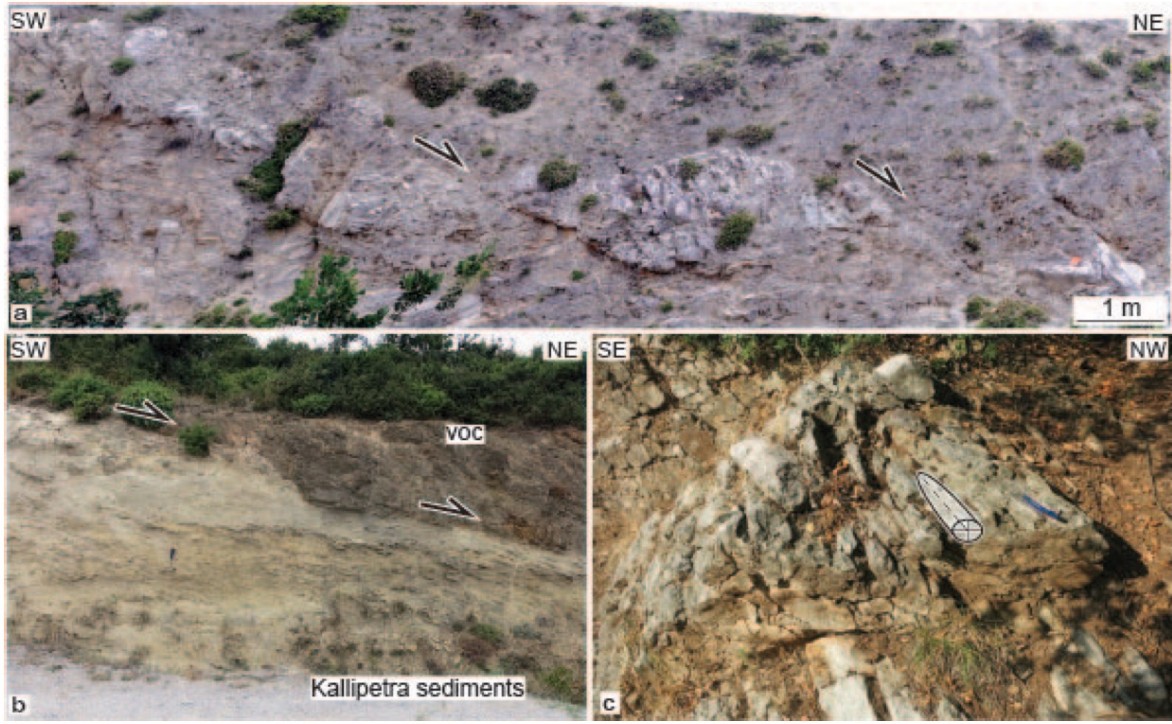

**Figure 7: (a) Asymmetric boudin of a limestone showing top-to-the NE shear sense. (b) Tectonic contact between the Kallipetra basin and the VOC with a top-to-the NE shearing. (c) Cigar-shaped clasts in a polymictic conglomerate of the mound top within the shear zone below the VOC (with sketch of the uniaxial ellipsoid mimicking the shape of the clasts). The stretch axis (X) of the prolate finite strain is parallel to the regional mineral and stretching lineation.**





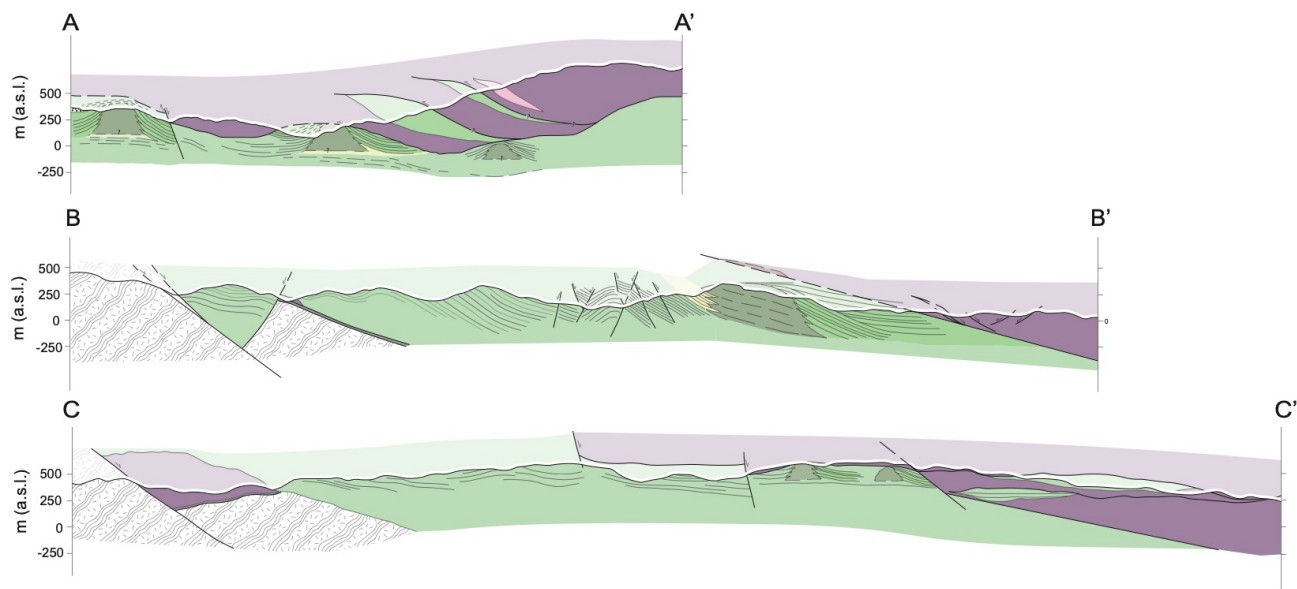


**Figure 8: Geologic cross-sections of the mapped area, colors corresponding to those on the geological map (Fig. 3).**



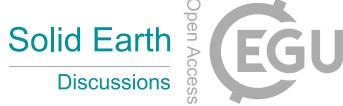

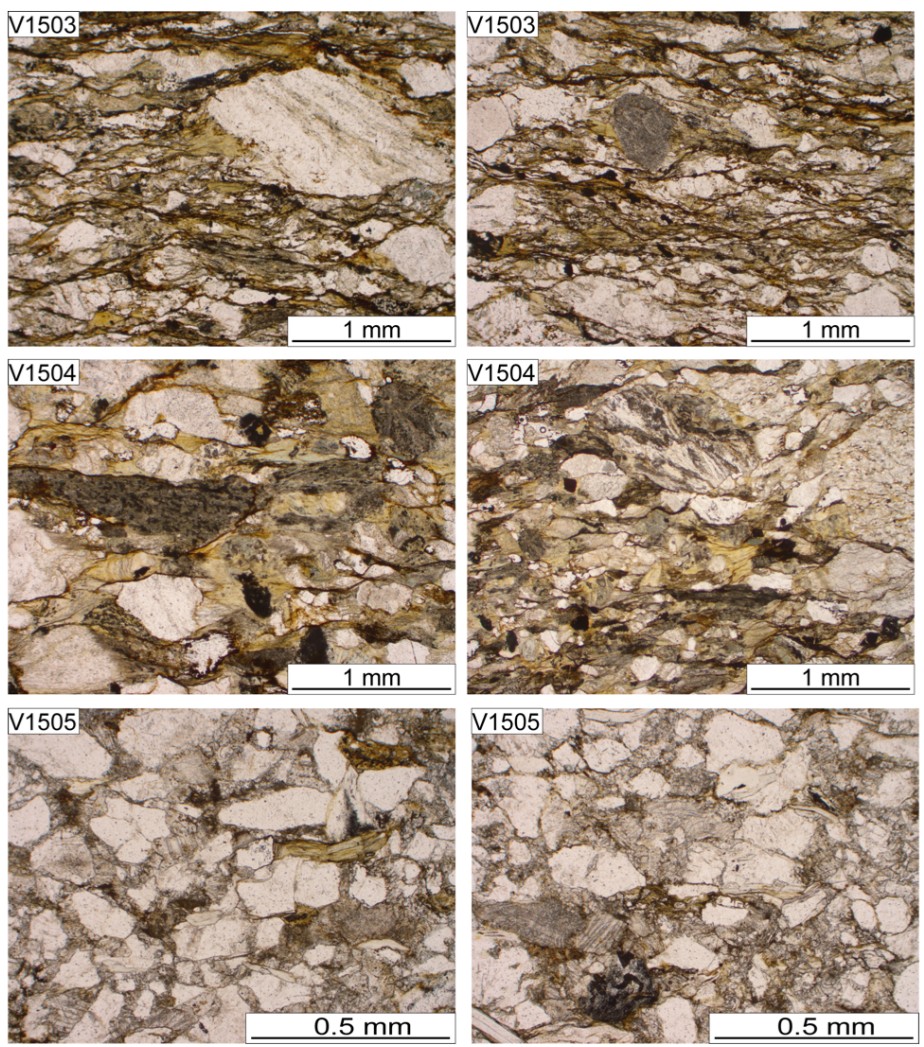

**Figure 9: (a) Sample V1503, newly formed chlorite; (b) Sample V1503, newly formed chlorite; (c) Sample V1504, newly formed chlorite; (d) Sample V1504, newly formed chlorite; (e) Sample V1505, detrital chlorite; (f) Sample V1505, detrital chlorite.**



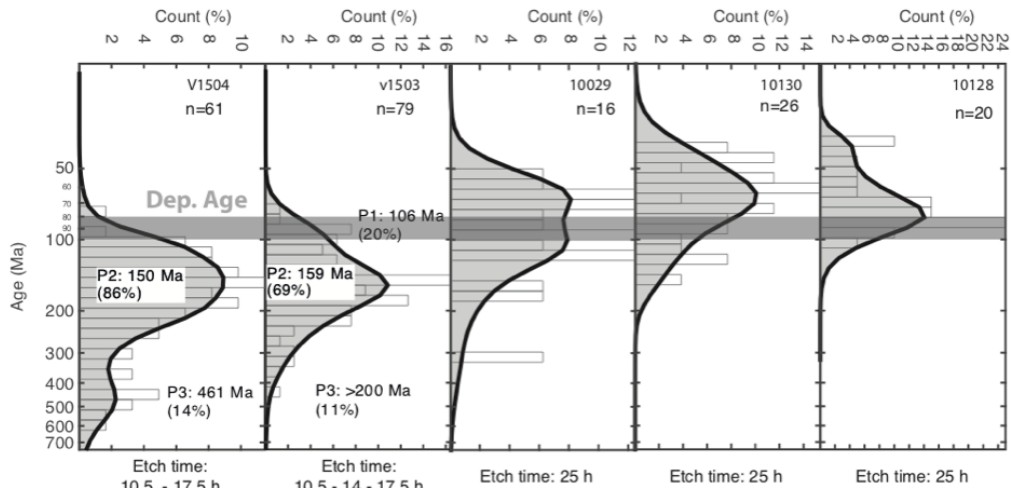

**Figure 10: Zircon fission track ages of samples taken in the mapped area.**

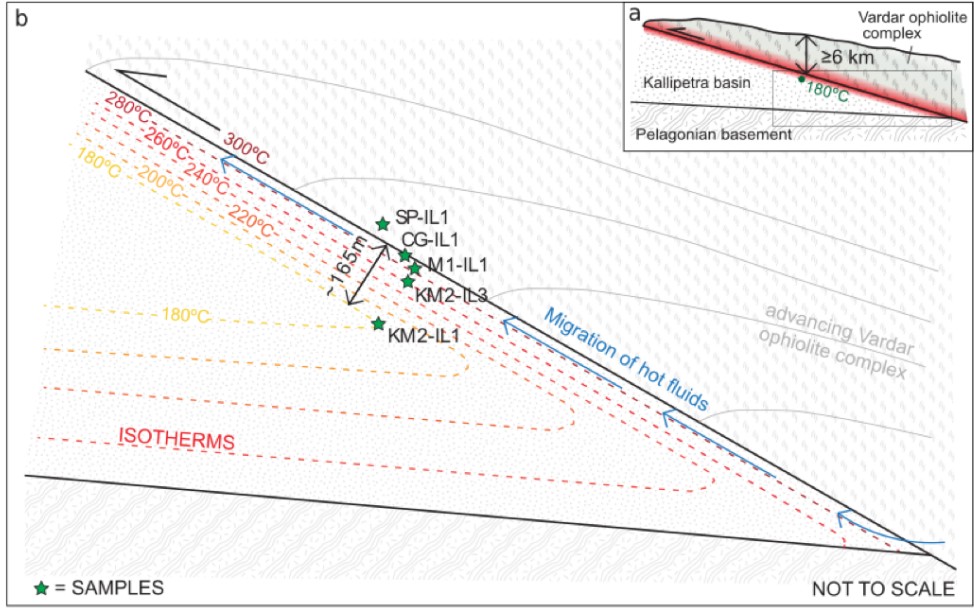


**Figure 11: Schematic diagram showing the inverse geothermal gradient at the contact between the VOC and Kallipetra Basin.**





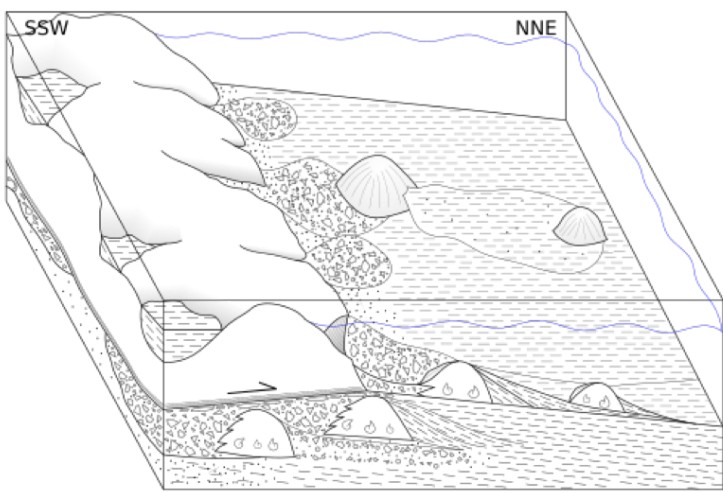

**Figure 12: Schematic diagram showing the sedimentary and tectonic environment of the Kallipetra Basin during the Turonian, and the overriding of rudist mounds by the resumed thrusting of the VOC.**



| Location | Foraminifera | Stratigraphic Distribution | Age |
|---|---|---|---|
| N 40º 28' 53" E 022º 13' 38" | *Helvetoglobotruncana helvetica* | Lower-Middle Turonian | 93.5 - 92.7 Ma |
| N 40º 28' 53" E 022º 13' 38" | *Dicarinella hagni (?)* | Lower Turonian - Coniacian | 93.5 - 86.7 Ma |
| N 40º 28' 53" E 022º 13' 39" | *Whiteinella* sp. | Upper Cenomanian-Campanian | 100.5 - 72.05 Ma |
| N 40º 28' 55" E 022º 13' 42" | *Whiteinella (inornata?)* | Cenomanian-Turonian boundary - Santonian | 94.03 - 84.19 Ma |
| N 40º 28' 55" E 022º 13' 42" | *Mesorbitolina pervia* | Mid - upper Aptian | |

**Table 1: Table of observed planktonic foraminifera.**




| Sample | Depth (m) | KI (AD) | KI (EG) | Metapelitic zone | Approx. T (°C) |
|---|---|---|---|---|---|
| SP IL1 | 540 | 0.091 | 0.109 | Epizone | 310 |
| CG IL1 | 535 | 0.141 | 0.122 | Epizone | 300 |
| M3/2 | 527 | no illite | | | 310 |
| M3/1 | 525 | no illite | | | 310 |
| M2/16 | 520 | 0.181 | | Low epizone | 295 |
| M2 IL3 | 519 | 0.126 | 0.11 | Epizone | 305 |
| M2/13 | 517 | 0.225 | | High anchizone | 280 |
| M2 IL2 | 515 | 0.258 | 0.131 | Low anchizone | 230 |
| M2/10 | 512 | 0.145 | | Epizone | 300 |
| M2/7 | 510 | 0.175 | | Epizone | 290 |
| M2/4 | 507 | 0.157 | | Epizone | 295 |
| M2/1 | 504 | 0.192 | | High anchizone | 290 |
| M1/7 | 500 | 0.137 | | Epizone | 300 |
| M1/5 | 498 | 0.131 | | Epizone | 300 |
| M1/3 | 497 | 0.22 | | High anchizone | 285 |
| M2 IL1 | 496.5 | 0.131 | 0.122 | Epizone | 300 |
| M1 IL1 | 496 | 0.127 | 0.116 | Epizone | 300 |
| M1/1 | 495 | 0.176 | | Epizone | 290 |
| CRN1/1 | 428 | 0.209 | | High anchizon | 285 |
| CRN1/3 | 418 | 0.25 | | Low anchizone | 275 |
| CRS1/2 | 350 | 0.18 | | High anchizon | 290 |
| CRS1/1 | 345 | 0.286 | | Low anchizone | 230 |
| CRS1/3 | 330 | 0.191 | | High anchizon | 290 |
| KM2 IL3 | 327 | 0.383 | 0.188 | Detrital | 250 |
| KM2 IL2 | 325 | 0.383 | 0.224 | Detrital | 250 |
| KM1/2 | 321 | 0.14 | | Smectite | 200 |
| KM1/1 | 320 | 0.168 | | | 200 |
| KM1 IL1 | 316 | 0.383 | 0.353 | Deep diagenetic zone | 160-200 |
| KM2 IL1 | 312 | 0.388 | 0.164 | Deep diagenetic zone | 160-200 |
| KM2/8 | 308 | 0.209 | | Diagenetic | 160-201 |
| KM2/6 | 304 | 0.166 | | Diagenetic | 160-202 |
| KM2/4 | 300 | 0.16 | | Diagenetic | 160-203 |
| KM2/2 | 298 | 0.14 | | Diagenetic | 160-204 |
| CRS1/4 | 230 | 0.286 | | Low anchizone | 230 |

**Table 2: Illite crystallinity data.**






| Sample ID | UTM | E | N | Elevation | Mount ID | Etch time | N. grains | $n_D$ | $\rho_D$ | $n_s$ | $\rho_s$ | $n_i$ | $\rho_i$ | $P\chi^2$ | Age Dispersion | Central Age | $\sigma 1$ |
|---|---|---|---|---|---|---|---|---|---|---|---|---|---|---|---|---|---|
| | | m | m | m | | hr | | tracks | e+05 tracks cm$^{-2}$ | tracks | e+06 tracks cm$^{-2}$ | tracks | e+06 tracks cm$^{-2}$ | % | % | Ma | Ma |
| V1503 | 34T | 607415.37 | 4476334.86 | 800 | a | 17.5 | 10 | 6594 | 5.290 | 5628 | 12.226 | 1353 | 2.939 | 0 | 24 | 155.99 | 10.07 |
| | | | | | b | 14 | 19 | 6579 | 5.279 | | | | | | | | |
| | | | | | c | 10.5 | 27 | 6565 | 5.267 | | | | | | | | |
| | | | | | d | 10.5 | 23 | 6551 | 5.256 | | | | | | | | |
| V1504 | 34T | 607415.37 | 4476334.86 | 800 | a | 17.5 | 18 | 6523 | 5.233 | 5101 | 13.396 | 1083 | 2.844 | 0 | 34 | 176.65 | 13.22 |
| | | | | | c | 10.5 | 43 | 6537 | 5.245 | | | | | | | | |

**Table 3: Zircon fission-track data. Variable amounts of zircons were analyzed on multiple mounts for each sample that were etched for different times. As a fluence monitor, a glass standard CN1 with a U concentration of 39.8 ppm was used. Central ages were calculated using a ζ calibration value of 145.39 ± 7.04. $n_D$ and $\rho_D$: number and density of induced tracks from the fluence monitor. $n_s$ and $\rho_s$: number and density of spontaneous tracks in the zircons. $n_i$ and $\rho_i$: number and density of spontaneous tracks from the zircons. $P_{\chi 2}$: $\chi 2$ probability.**