# Peer review of "Birth and closure of the Kallipetra Basin: Late Cretaceous reworking of the Jurassic Pelagonian – Axios-Vardar contact (Northern Greece)"

_Solid Earth, 2020_

## Referee Comment (RC1) · Anonymous Referee #1 · 13 Aug 2020

Dear Editor, I have read with great interest the work of Bailey and coauthors regarding the Kallipetra Basin in N. Greece. The manuscript is well written and it presents new data and interpretations in connection to the geodynamics and the problems of ophiolite obduction in the Hellenides. The writing is clear and easy to follow. The authors have put a great effort to document the data and the field evidence related to this study and I have to admit that it is rare to see papers with such a level of detail when it comes to the primary data. However, I have a few comments related to the overall presentation and some of the conclusions of the study. The most important points that I can mention here are: 1) The focus of the study with respect to the general problematics of tectonic scenarios in the Hellenides must be addressed more clearly. This is because

the study may look a bit "too regional" from the perspective of a researcher who is not familiar with Eastern Mediterranean geology. 2) The authors present their view of the main contact being a tilted thrust towards the NNE. This is a very important point given the current discussion in the literature related to the ophiolite obduction problem (Pindos vs Vardar etc). I would thus suggest that the authors develop the discussion and explain their reasoning a bit in more detail. 3) The authors get into an extensive discussion about the importance of fluids vs viscous heating while at the same time they also admit that the evidence is not so clear. Since the discrimination of the additional source of heat which is required is not the topic of this study (and there has been no effort in quantify the arguments), I would suggest that the authors mention the possibilities and not go in a specific discussion on the importance of a particular mechanism.

The Specific comments follow below:

l. 27: Please add "e.g." in the reference list. There are numerous works to be cited here.

l. 28: Please define what is meant by "Internal" Hellenides, either by definition or by citation to the map.

l. 34: Please be more specific about the "Cretaceous Basin". Does it have a name? Is there in a particular location that you refer to?

l. 37: I would suggest that "pulse" is not the right word here. It is known that the extension and basin formation in the Hellenides is diachronous and migrating southwards (see also Papanikolaou & Royden, 2007) for more details and the relevant literature.

l. 41 (MAIN POINT): It is not clear what are the main features that you would like to address in all these contrasting interpretations. In terms of the sketches that are presented in Fig. 2 the focus of this work can be i) the position of Pelagonia, ii) the number of subduction zones etc. Therefore, I suggest you develop on the specific features that you want to address in more detail. In other words, please identify the

problem/hypothesis and then explain why you chose to focus on this area to solve it/ test it.

l. 68: Please add e.g. in the citation list since the development of these basins were known already from the time of Brunn and Aubouin (1950-60s)

l. 77: "metamorphic ages of migmatites" should change to "zircon ages from the leucosomes from the migmatites".

l. 80: "of the wedge" Please rephrase so that you can be more specific on the kind of the wedge (e.g. accretionary, orogenic etc).

l. 93: Please add Brun & Sokoutis as well as Dinter & Royden for the Rhodope core complexes.

l. 95: leucogneiss -> leucogneisses

l. 105: Please avoid terms that refer to processes which you cannot show (i.e. "hydraulically").

l. 117: Please add reference to show who did this interpretation (after "basin").

l. 118: As above, please add reference at the end.

l. 127: "package" -> "pile"?

l. 131: Please be specific because there are also other kinds of grade (i.e. ore grade). I suggest rewording as: "to determine grade. . ." -> to determine the metamorphic grade in low-grade metapelitic . . .

l. 141: What exactly do you mean by the "determination of metapelitic zones". I think you refer to the "metamorphic" zones. Right?

l. 168: As before, please remote the word "hydraulically".

l. 449: "dramatic" has been struck through. Please check the sentence.

l. 457: "and on viscosity". I would remove the specific mention to "and on viscosity" since any irreversible deformation mechanism would also contribute to shear heating (e.g. rate-independent plasticity)

l. 460: "With a <2cm/a the heat is…" This statement assumes that the movement is steady. Since this hypothesis cannot be supported by the present data, I would suggest removing this sentence.

l. 465-467 (MAIN POINT): As before, the discussion around viscosity only, neglects the frictional part of the heat. Therefore, since this is not the main topic of this paper and there is no detailed analysis in this direction, I would remove specific conclusions related to the most-likely source and the magnitude of shear heating.

l. 470-471: As before, there is no evidence to suggest what is considered "normal" by the authors since: (i) the rheology does not have to be purely viscous, (ii) the motion does not need to be steady. Therefore, the suggestion of a particular heating mechanism is beyond the scope and the data presented in this study.

l. 475: Why the direction of tectonic transport is related to the fluid flow. Assuming a fault zone as a region of high permeability is well established. However, I cannot see how the transport is related for this conclusion.

l. 488-490: How did you conclude that this must be thrusting (MAIN POINT). Why not normal fault with top NE kinematics. Please explain your reasoning in more detail. l. 494: As before, since the authors already state in line 478 that the sources of heat are not clearly established. I would leave the interpretations out of this.

l. 494-496: These places are quite far from each other.

l. 496-498: From Turonian to Campanian is more than 10Myr. For a crust ∼10km thick and standard thermal parameters, the conductive thermal relaxation timescale is ca 1Myr. Therefore, I do not think that the advective heat was maintained long enough to cause the heating. Therefore, I would suggest that the authors revise this sentence to

defend or reject this conclusion.

Legend Fig. 3: "Dark blue circles", the samples are very small. Please use larger and more discrete symbols.

---

## Referee Comment (RC2) · Anonymous Referee #2 · 19 Aug 2020

This work deals with the paleogeographic and tectonic evolution during the Upper Cretaceous of an area of Continental Greece that belongs to the so-called Internal Hellenides. Little is known about the Cretaceous evolution of this sector of the Hellenides and many questions await answers. Apart from the number of oceanic basins, the polarity of the subduction zone, etc, there are questions about the origin, age, deposition paleoenvironment and the geodynamic significance of the Cretaceous sediments deposited unconformably on top of the obducted Vardar ophiolite complexes and the Pelagonian passive margin. Thus, this manuscript fills a significant gap in our knowledge of these issues.

[Figure]

It is a well-written and well-structured manuscript with a wealth of data clearly presented, but in the end, it leaves the reader partially dissatisfied. And this has to do mainly with the large-scale implications of the results and their comparison with other neighboring areas of the Internal Hellenides where Upper Cretaceous sediments are also observed. As the authors report, in order to elucidate part of the controversies, they studied this small Upper Cretaceous basin, but the part of their manuscript that refers to those is poorly developed. I believe that a better analysis of this would strengthen their work even more.

Based on that I have noted the following:

a) The work that first described the Cretaceous sediments east of the Pelagonian (Almopias Zone) is not included in the reference list, although this work is about an area just north of the Kallipetra basin and gives detailed lithostratigraphic columns presenting their paleogeographic and tectonic evolution. This work is:

Mercier, J., 1968. Etude geologique des zones Hellenides en Macedoine centrale (Grece). Ann. Geol. Pays Hell. 20 (792 pp.).

b) There is no comparison or correlation with other areas where the Cretaceous sediments are also observed. There could be a comparison apart from Mercier's work with the results of other papers, e.g. the paper of Sharp and Robertson (2006), who give an evolution model of a similar Upper Cretaceous basin north of the study area. Mercier (1968) places the beginning of the deposition of the Upper Cretaceous sediments in Aptian-Albian, while other researchers such as Sharp and Robertson and Galeos et al. (1994) describe even older aged sediments (Upper Jurassic). It could also be compared to other areas of the non-metamorphic Pelagonian, e.g. in Othrys Mt (Ferriere, 1982) and Argolida (Baumgartner, 1985). It is important to comment on the age of onset of the deposition of the Upper Cretaceous sediments, as well as the age of the emplacement of ophiolitic complexes on them, highlighting the possible differences that may exist from region to region.

Ferriere J (1982) Paleogeographies et tectoniques superposees dans les Hellenides Internes au niveau de l'Othrys et du Pelion (Grèce). Soc Geol Nord Publ 8:1–970.

Galeos, A., Pomoni-Papaioannou, F., Tsaila-Monopolis, S., Turnsek, D. & Ioacim, C. 1994. Upper Jurassic–Lower Cretaceous 'molassic-type' sedimentation in the western part of the Almopia subzone, Aridhea Loutra Unit (northern Greece). 7th Congress of the Geological Society of Greece, Thessaloniki, May 1994.

c) The phrase "Upper Cretaceous basin" is used in two ways: either to describe the wider paleogeographic area where the Upper Cretaceous sediments were deposited or the small basin of Kallipetra. This dual use of the term confuses the reader. It must be made clear that the Kallipetra basin is part of a wider paleogeographic domain which, during the Upper Cretaceous, was the site of deposition of large thickness sediments.

d) An important key in the evolution of the basin is the origin of the fault that places the Vardar Oceanic Complexes (VOC) on the Upper Cretaceous sediments in the eastern part of the basin. According to the authors, the direction of tectonic transport of the VOC sealing Kallipetra Basin was from SSW to the NNE. It seems difficult that this transport can place the VOC on the sediments of the basin in a distance at least 4 km into the basin and westwards, as shown by the geological map in Figure 3 and the geological sections in Figure 8. This could happen if the VOC nappe crossed the entire basin from southwest to northeast. Also, in the map of figure 3 the fault is characterized as a reactivated thrust fault. This is not clearly described in the text except perhaps from the sentence in line 490. A much better analysis and documentation of the interpretation given is needed.

e) What is the origin of the basin and how is it associated with the growth of the Hellenides? Is it a fore-arc basin formed on top of an evolving accretionary wedge, is it a basin formed at the back of an orogenic wedge that collapsed due to underplating at its base, or is it a back-arc basin?

f) The evolution of the basin could be captured by a series of sketches, which can be

either NE-SW striking cross-sections or 3D sketches, beyond the snapshot of Figure 12.

Comments on the text of the manuscript:

Line 28: There are dozens of references that could be placed here. It is better to include "e.g." at the beginning of the reference list.

Line 28: You should give the definition for the Internal Hellenides as the term is not only geographical or spatial but also has a geodynamic meaning by dividing the Hellenides into two areas with different evolution during the alpine orogenesis. Also, the first letter must be uppercase (Internal).

Line 34: What is the origin of this "Upper Cretaceous basin"? How was it created? Is it a single basin or more?

Line 36: I think that the migration is towards the SW-SSW.

Line 41: What are these controversies? I believe it needs further analysis beyond a simple reference to "controversies" and the presentation of a figure (Figure 2). You need to clarify the problem that you want to solve with this work.

Line 68: Add "e.g." at the beginning of the reference list as there are numerous works that could be cited here.

Lines 78-81: The area in which this stratigraphic gap has been described (Aptian-Albian) is very far from the study area and paleogeographically belongs to the western margin of Pelagonian and not to the eastern. Furthermore, other researchers (e.g. Sharp and Robertson 2006) argue that the onset of sedimentation occurs during the Aptian-Albian north of the study area.

Line 82: There are papers that describe older in age transgressive sediments which unconformably overlay the Pelagonian and Vardar units (e.g. Mercier 1968; Brown and Robertson 2004; Sharp and Robetson 2006; etc). See also my comment b.

[Figure]

Line 83: You need to add more references here. There are numerous works to be cited here, with primary data except from the synthetic work of Papanikolaou (2009).

Line 92: Add "e.g." at the beginning of the reference list as there are numerous works that could be cited here.

Line 95: Leucogneisses?

Lines 95-96: Are you referring exclusively to the area west of the Kallipetra Basin or to the Pelagonian in general? If the latter is true you should add more references, as it is not only Schenker (2013) who describes the above lithologies. You could add "Schenker 2013 and references therein".

Line 111-112: The sentence Âńthe sediments belong. . . . . . . .. . ..as the Kallipetra basinÂż causes confusion (see also previous comment c). What is called as Kallipetra basin? Is it the paleogeographic domain where the large thick Upper Cretaceous sediments were deposited or only the small basin under study?

Lines 115-118: Please enter references as you seem to be referring to older works.

Line 141: What do you mean by the term "metapelitic zones"?

Lines 235-236: You argue that the fossils are deformed and reworked and are supplied by the VOC based only on the work of Schenker (2013). Apart from this study, I do not remember any other study that reports Lower Cretaceous sediments in the VOC. On the contrary, there are papers that support the start of deposition in Aptian-Albian (see also previous comment b). Even in your own work it is described that sediments of Kallipetra Formation with VOC form duplexes, so how are you convinced that the fossils belong to VOC and not to the Kallipetra formation? Îďhere are also studies that describe Upper Jurassic-Lower Cretaceous sediments unconformably on the VOC, which seal the tectonic emplacement of the VOC onto the passive margin of the Pelagonian. If you include those Upper Jurassic-Lower Cretaceous sediments in what you have named as Vardar Oceanic Complexes then you need to clarify that.

Lines 311 and 312: Please correct the references. There is no Schenker (2014) in your reference list.

Line 415: Please enter reference as you seem to be referring to older work.

Line 449: The word "dramatic" has been struck through. I believe you need to delete that word.

Lines 488-489: See my comment d. As in the following lines (490-492) you suggest a localized inversion that predated the start of the general convergence, you have to enforce your interpretation.

Lines 494-498: I suggest to delete this interpretation as you have already weakened it in the second sentence.

Comments on the Figures

1. Figure 2 shows various models of evolution of the Hellenides in the Cretaceous, which are not analyzed in the manuscript and in the end there isn't any suggestion about them. Therefore, it does not offer anything substantial to this work and could be removed.

2. In the geological map of figure 3 some things are not visible and difficult to distinguish, e.g. difficult to distinguish black dots from dark blue ones. Therefore, some symbols need to be magnified.

3. In the geological sections of Figure 8, there is a large number of faults. According to the manuscript and the map of figure 3, these are normal, thrust and strike- slip faults. In order for the reader to find out which is which, he must constantly resort to the map. Therefore, I suggest the relative slip of the fault-blocks should be plotted along the faults.

4. In figure 12 there is no legend explaining the symbols used to describe the different geological formations of the sketch. The sketch also gives a false impression that the

basin has developed mainly east and northeast of the VOC. Perhaps the sketch should also include the western margin of the basin in order for the reader to have a complete picture. See also previous comment for 3D sketches.

---

## Author Comment (AC1) · 11 Sep 2020

We are grateful for the positive response and constructive comments provided by the anonymous referee. They raise 3 main points: (1) the focus of the study needs to be addressed more clearly in the introduction and in the discussion/conclusions; (2) our discussion and reasoning on the tilted thrust fault should be developed further; (3) the extensive discussion on the origin of fluids in the fault zone is not the topic of this study. These are addressed below, and specific comments to individual points of the manuscript are provided. We will upload a revised manuscript and edited/new figures shortly.

[Figure]

Anonymous Referee #1 Dear Editor, I have read with great interest the work of Bailey and coauthors regarding the Kallipetra Basin in N. Greece. The manuscript is well written and it presents new data and interpretations in connection to the geodynamics and the problems of ophiolite obduction in the Hellenides. The writing is clear and easy to follow. The authors have put a great effort to document the data and the field evidence related to this study and I have to admit that it is rare to see papers with such a level of detail when it comes to the primary data. However, I have a few comments related to the overall presentation and some of the conclusions of the study. The most important points that I can mention here are: 1) The focus of the study with respect to the general problematics of tectonic scenarios in the Hellenides must be addressed more clearly. This is because the study may look a bit "too regional" from the perspective of a researcher who is not familiar with Eastern Mediterranean geology.

We will alter the introduction of our manuscript to more clearly address the focus of our study with respect to the 'controversies' of the Hellenides, please see our more detailed response on this issue below, in the response to a comment on l.41 by the referee.

2) The authors present their view of the main contact being a tilted thrust towards the NNE. This is a very important point given the current discussion in the literature related to the ophiolite obduction problem (Pindos vs Vardar etc). I would thus suggest that the authors develop the discussion and explain their reasoning a bit in more detail.

We thank the reviewer for this suggestion. We will add a sketch for more clear presentation of our tilted thrust zone and discuss our reasoning in more detail in the manuscript.

3) The authors get into an extensive discussion about the importance of fluids vs viscous heating while at the same time they also admit that the evidence is not so clear. Since the discrimination of the additional source of heat which is required is not the topic of this study (and there has been no effort inquantify the arguments), I would suggest that the authors mention the possibilities andnot go in a specific discussion on the importance of a particular mechanism.

Our data document an inverted metamorphism below a shear zone. The heating that produced this metamorphism occurred during deformation and reset the FT ages, permitting us to date deformation. We acknowledge that physical models are needed to reproduce and quantify length and timing of the observed metamorphism. However, our field observations help to rule out some processes related to the different possible heat sources. We therefore think that the critical issue of heat source should remain in this study. However, as asked by the reviewer, will reduce the extensive discussion so the reader can simply recognize the presence of heat along the thrust zone without getting deterred from the main conclusions of our study.

The Specific comments follow below: l. 27: Please add "e.g." in the reference list. There are numerous works to be cited here.

Done.

l. 28: Please define what is meant by "Internal" Hellenides, either by definition or by citation to the map.

We will refer to the map and edit the map (Fig. 1) accordingly.

l. 34: Please be more specific about the "Cretaceous Basin". Does it have a name? Is there in a particular location that you refer to?

Here we refer to the Kallipetra Basin. The Cretaceous basins that formed at the eastern Pelagonian margin and over the Axios/Vardar zone were first mapped at the large scale by Kossmat 1924. Many workers have found sparse Cretaceous sediments since then (e.g. Mercier and Vergely 1994, Sharp and Robertson 2006). Schenker et al (2015) brought clear evidence that the gneissic detritus in one of these basins (named in this contribution Kallipetra basin) is of Pelagonian and not of Rhodopian origin (e.g. Ricou and Godfriaux 1995). We will rephrase this part of the text: "...by the deposition of metamorphic Pelagonian detritus in a Late Cretaceous basin (Schenker et al. 2015) subsequently referred to as the Kallipetra Basin in this study."
l. 37: I would suggest that "pulse" is not the right word here. It is known that the extension and basin formation in the Hellenides is diachronous and migrating southwards (see also Papanikolaou & Royden, 2007) for more details and the relevant literature.

We will rephrase to: "Finally, from the Oligocene-Miocene the western Pelagonia was dissected by diachronous normal faults (Schermer et al., 1990; Lacassin et al., 2007; Coutand et al., 2014; Schenker et al., 2014) within a southward extensional deformation front that affected most of the Hellenides (e.g. Papanikolaou & Royden, 2007)."

l. 41 (MAIN POINT): It is not clear what are the main features that you would like to address in all these contrasting interpretations. In terms of the sketches that are presented in Fig. 2 the focus of this work can be i) the position of Pelagonia, ii) the number of subduction zones etc. Therefore, I suggest you develop on the specific features that you want to address in more detail. In other words, please identify the problem/hypothesis and then explain why you chose to focus on this area to solve it/test it.

The many and contrasting geodynamics models present in the literature source from the difficulties of connecting the Rhodope and the Pelagonian zone. This is a long-standing debate on the number and dimension of oceans in the Mesozoic Pindos-Vardar realm between researchers proposing a single unifying Early Jurassic Vardaric ocean that has been partly subducted, partly obducted and dismembered during successive tectonic events and researchers that embraced a multi-ocean early Jurassic scenario that led to several subduction zones. In these scenarios, the Cretaceous sediments were deposited on the eastern Pelagonian zone either within a Jurassic-Cretaceous passive margin or during a subsequent Cretaceous tectonic event (compressional or extensional depending on the authors). These geodynamic interpretations are presented in Fig.2 and display the different positions of the Pelagonia-Vardar margin relative to the Alpine orogenic wedge after the Jurassic convergence. By studying the small Upper Cretaceous Kallipetra Basin that lies on the Pelagonia-Vardar 'suture zone', we can begin to address questions on if and how the Pelagonian-Vardar

margin was deforming. Our study will ultimately provide constraints on the position of the eastern Pelagonian margin relative to the Alpine orogenic wedge, hence ruling out some of the geodynamic models so far proposed. Thanks to the comments of both reviewers, we will adjust the manuscript so as to better identify the problem we want to address, and to show the importance of the birth and the closure of the Kallipetra basin in the context of the Hellenides.

l. 68: Please add e.g. in the citation list since the development of these basins were known already from the time of Brunn and Aubouin (1950-60s)

Done.

l. 77: "metamorphic ages of migmatites" should change to "zircon ages from the leucosomes from the migmatites".

OK, we agree: the term proposed is more descriptive.

l. 80: "of the wedge" Please rephrase so that you can be more specific on the kind of the wedge (e.g. accretionary, orogenic etc).

We mean orogenic wedge and will rephrase accordingly.

l. 93: Please add Brun & Sokoutis as well as Dinter & Royden for the Rhodope corecomplexes.

This can be done.

l. 95: leucogneiss -> leucogneisses

Done.

l. 105: Please avoid terms that refer to processes which you cannot show (i.e. "hydraulically").

The unit "hydraulically brecciated serpentinite" will be re-named to "Dark massive fractured to brecciated serpentinites".

l. 117: Please add reference to show who did this interpretation (after "basin").

OK, it is the interpretation of Schenker et al 2015.

l. 118: As above, please add reference at the end.

Schenker et al 2015

l. 127: "package" -> "pile"?

We will replace "sedimentary package" with "sedimentary sequence".

l. 131: Please be specific because there are also other kinds of grade (i.e. ore grade). I suggest rewording as: "to determine grade..." -> to determine the metamorphic grade in low-grade metapelitic

We agree with this suggestion, and the phrase "to determine grade" will be replaced with "to determine diagenetic grade".

l. 141: What exactly do you mean by the "determination of metapelitic zones". I think you refer to the "metamorphic" zones. Right?

Yes, we refer to low-grade metamorphic zones, so we will replace 'metapelitic zones' with 'low grade metamorphic zones' in the manuscript.

l. 168: As before, please remote the word "hydraulically".

We will remove the term 'hydraulically' and use "'Dark massive fractured to brecciated serpentinites", as mentioned in an above comment.

l. 449: "dramatic" has been struck through. Please check the sentence.

Thank you for alerting us to this, the word dramatic should be removed.

l. 457: "and on viscosity". I would remove the specific mention to "and on viscosity"since any irreversible deformation mechanism would also contribute to shear heating (e.g. rate-independent plasticity)

We agree with the reviewer and deleted "and on viscosity".

l. 460: "With a <2cm/a the heat is..." This statement assumes that the movement is steady. Since this hypothesis cannot be supported by the present data, I would suggest removing this sentence.

See comment below the following point.

l. 465-467 (MAIN POINT): As before, the discussion around viscosity only, neglects the frictional part of the heat. Therefore, since this is not the main topic of this paper and there is no detailed analysis in this direction, I would remove specific conclusions related to the most-likely source and the magnitude of shear heating.

As the reviewer has helpfully pointed out, the discussion around the specific magnitudes of heating related either to shear heating or advected hot fluids is highly hypothetical, and we therefore do not have adequate evidence to support one of the two sources of heat. Rather, the goal of this particular paragraph was to draw attention to the unusual inverse geothermal gradient and explore possibilities of how/why this formed. Therefore, we will re-write, simplify, and shorten Section 5.5 'The inverted geothermal gradient in the Kallipetra Basin' to address the concerns of Referee #1.

l. 470-471: As before, there is no evidence to suggest what is considered "normal" by the authors since: (i) the rheology does not have to be purely viscous, (ii) the motion does not need to be steady. Therefore, the suggestion of a particular heating mechanism is beyond the scope and the data presented in this study.

We acknowledge that we have no evidence or data that addresses the convergence rates, viscosities, or plate velocities and therefore agree that the suggestion of particular heating mechanisms goes beyond the scope and data presented in this study. Therefore, we will re-write and shorten this section so that we only relate the observed inverse geothermal gradient to the closure of the Kallipetra Basin so that it remains in the scope of our study.

l. 475: Why the direction of tectonic transport is related to the fluid flow. Assuming a fault zone as a region of high permeability is well established. However, I cannot see how the transport is related for this conclusion.

We agree that transport is not necessarily related to this conclusion, therefore we will replace "The overriding unit over the Kallipetra basin would have allowed fluid focusing and differential loading that caused any fluids to flow in the direction of tectonic transport" with "Differential loading from the overriding unit over the Kallipetra Basin could have focused fluids along the fault zone".

l. 488-490: How did you conclude that this must be thrusting (MAIN POINT). Why not normal fault with top NE kinematics. Please explain your reasoning in more detail.

The conclusion for thrusting to the NE came from the stratigraphic evidence, predominantly from the character of the rudist mounds. We see stacking of serpentinitic breccias on south-western flanks of rudist mounds, sourced from ophiolitic debris up slope. The absence of ophiolitic detritus on the northeastern mound flanks document a 'shadow' effect of the mounds with respect to a south-southwestern provenance of serpentinite clasts. The highest, and therefore youngest, mound is located at Asomata Quarry which is the most northeastern mound suggesting movement of the ophiolite from SSW to NNE. Part of the reason the rudist mounds are so interesting is that they tell us something about the tectonic activity the basin is experiencing without needing to observe the fault itself. We will add a 2D sketch that documents this evolution that should make the reasoning clearer for the reader.

l.494: As before, since the authors already state in line 478 that the sources of heat are not clearly established. I would leave the interpretations out of this.

See comments above pertaining to this issue. We will remove the interpretations of heat sources from this sentence.

l. 494-496: These places are quite far from each other.

Indeed they are. We will delete this sentence.

l. 496-498: From Turonian to Campanian is more than 10 Myr. For a crust âĹij10km thick and standard thermal parameters, the conductive thermal relaxation timescale is ca1Myr. Therefore, I do not think that the advective heat was maintained long enough to cause the heating. Therefore, I would suggest that the authors revise this sentence to defend or reject this conclusion.

We will also delete this sentence as it directly follows from the previous sentence which was deleted in response to the reviewer comment above.

Legend Fig. 3: "Dark blue circles", the samples are very small. Please use larger and more discrete symbols.

We agree, very small - we will adjust the figure.

Please also note the supplement to this comment:
https://se.copernicus.org/preprints/se-2020-106/se-2020-106-AC1-supplement.pdf

---

## Author Comment (AC2) · 11 Sep 2020

We thank the reviewer for their detailed and thorough review of the manuscript, which will allow us to significantly improve it. The referee's main point was addressing the lack of large-scale implications and comparison with neighboring areas with Upper Cretaceous sediments. We have addressed their main concerns and respond to their individual comments below. A revised manuscript along with edited and newly created figures will be uploaded shortly, which will include further comparison to nearby regions and a clearer statement of our study goals and the controversies we wish to elucidate.

Anonymous Referee #2

[Figure]

This work deals with the paleogeographic and tectonic evolution during the Upper Cretaceous of an area of Continental Greece that belongs to the so-called Internal Hellenides. Little is known about the Cretaceous evolution of this sector of the Hellenides and many questions await answers. Apart from the number of oceanic basins, the polarity of the subduction zone, etc, there are questions about the origin, age, deposition paleoenvironment and the geodynamic significance of the Cretaceous sediments deposited unconformably on top of the obducted Vardar ophiolite complexes and the Pelagonian passive margin. Thus, this manuscript fills a significant gap in our knowledge of these issues. It is a well-written and well-structured manuscript with a wealth of data clearly presented, but in the end, it leaves the reader partially dissatisfied. And this has to do mainly with the large-scale implications of the results and their comparison with other neighboring areas of the Internal Hellenides where Upper Cretaceous sediments are also observed. As the authors report, in order to elucidate part of the controversies, they studied this small Upper Cretaceous basin, but the part of their manuscript that refers to those is poorly developed. I believe that a better analysis of this would strengthen their work even more.

Based on that I have noted the following: a) The work that first described the Cretaceous sediments east of the Pelagonian (Almopias Zone) is not included in the reference list, although this work is about an area just north of the Kallipetra basin and gives detailed lithostratigraphic columns presenting their paleogeographic and tectonic evolution. This work is: Mercier, J., 1968. Etude geologique des zones Hellenides en Macedoine centrale(Grece). Ann. Geol. Pays Hell. 20 (792 pp.).

We will enter in more detail with the comparison of the Lower Cretaceous basin referring to the work of Mercier, Robertson, Ricou and others. We hope that by adding some detailed comparisons with other Upper Cretaceous basins in nearby regions will strengthen the part of our manuscript that aims to elucidate the controversies.

b) There is no comparison or correlation with other areas where the Cretaceous sediments are also observed. There could be a comparison apart from Mercier's work

with the results of other papers, e.g. the paper of Sharp and Robertson (2006), who give anevolution model of a similar Upper Cretaceous basin north of the study area. Mercier(1968) places the beginning of the deposition of the Upper Cretaceous sediments in Aptian-Albian, while other researchers such as Sharp and Robertson and Galeos etal. (1994) describe even older aged sediments (Upper Jurassic). It could also be compared to other areas of the non-metamorphic Pelagonian, e.g. in Othrys Mt (Ferriere,1982) and Argolida (Baumgartner, 1985). It is important to comment on the age of onset of the deposition of the Upper Cretaceous sediments, as well as the age of the emplacement of ophiolitic complexes on them, highlighting the possible differences that may exist from region to region.

We agree with this point (see reply above) and we will compare/correlate our units with others, specifically with ages of deposition and/or emplacement. We will be sure to read the literature stated below by the reviewer to compare to our work.

Ferriere J (1982) Paleogeographies et tectoniques superposees dans les HellenidesInternes au niveau de l'Othrys et du Pelion (Grèce). Soc Geol Nord Publ 8:1–970.

Galeos, A., Pomoni-Papaioannou, F., Tsaila-Monopolis, S., Turnsek, D. & Ioacim, C.1994. Upper Jurassic–Lower Cretaceous 'molassic-type' sedimentation in the westernpart of the Almopia subzone, Aridhea Loutra Unit (northern Greece). 7th Congress of the Geological Society of Greece, Thessaloniki, May 1994.

c) The phrase "Upper Cretaceous basin" is used in two ways: either to describe the wider paleogeographic area where the Upper Cretaceous sediments were deposited or the small basin of Kallipetra. This dual use of the term confuses the reader. It must be made clear that the Kallipetra basin is part of a wider paleogeographic domain which, during the Upper Cretaceous, was the site of deposition of large thickness sediments.

Thank you for bringing this to our attention - we will go through the manuscript to make sure this is cleared up to eliminate any confusion. In line 34, for example, we refer to the Kallipetra Basin and will rephrase this part of the text: "...by the deposition of

metamorphic Pelagonian detritus in a Late Cretaceous basin (Schenker et al. 2015) subsequently referred to as the Kallipetra Basin in this study", and be sure to make clarify other references to 'Upper Cretaceous Basin' in our manuscript.

d) An important key in the evolution of the basin is the origin of the fault that places the Vardar Oceanic Complexes (VOC) on the Upper Cretaceous sediments in the east-ernpart of the basin. According to the authors, the direction of tectonic transport of the VOC sealing Kallipetra Basin was from SSW to the NNE. It seems difficult that this transport can place the VOC on the sediments of the basin in a distance at least 4km into the basin and westwards, as shown by the geological map in Figure 3 and the geological sections in Figure 8. This could happen if the VOC nappe crossed the entire basin from southwest to northeast. Also, in the map of figure 3 the fault is char-acterized as a reactivated thrust fault. This is not clearly described in the text except perhaps from the sentence in line 490. A much better analysis and documentation of the interpretation given is needed.

We do not fully understand the argument in this comment, however this, along with a similar comment from Reviewer 1, has alerted us to confusion over the reactivated thrust fault and direction of transport in our manuscript. We will make sure there is a better description and documentation of the reactivated thrust fault in section 5.6 'Sealing of the Kallipetra Basin and large-scale implications'. We will also add a series of 2D sketches that shows (1) the north eastward migration of the mounds is related to thrusting and not to normal faulting, (2) normal vs. inverted thrust, and (3) subsequent rotation of the fault into a 'normal' position.

e) What is the origin of the basin and how is it associated with the growth of the Hel-lenides? Is it a fore-arc basin formed on top of an evolving accretionary wedge, is it a basin formed at the back of an orogenic wedge that collapsed due to underplating at its base, or is it a back-arc basin?

Towards the end of the Kallipetra Basin timeline, the basin could be described as sed-

iments accumulating in a foredeep generated ahead of an emplacing ophiolite. However, the basin formed under an extensional exhumation phase where there was a lot of erosion of both the Pelagonian continent and the Jurassic ophiolite, forming a depression. The upward deepening of the successions (before we shallow again due to incoming ophiolite), suggests a phase of extension. Sharp & Robertson (1993) document a phase of extension affecting much of the Vardar Zone and other parts of the Hellenides in the Turonian - however this is when we document closure and compression in the Kallipetra. This brings further attention to an earlier important point raised by Referee #2 - that an analysis and comparison of our study site with Cretaceous Basins from other works will strengthen our study and highlight the heterogeneity that exists from region to region. We will build this information into the conclusions/interpretations of the Kallipetra Basin and how it was formed. We had (1) Jurassic compression and ophiolite emplacement; (2) Re-opening under transtension/extension early-mid Cretaceous time; and (3) Closure and thrust fault reactivation in Late Cretaceous (and 'suture zone tightening').

f) The evolution of the basin could be captured by a series of sketches, which can be either NE-SW striking cross-sections or 3D sketches, beyond the snapshot of Figure12.

We think this will be a good addition to the manuscript that might help solidify some of our explanations and interpretations, especially with regards to your point (d). We will replace figure 12 with a snapshot of various times: (1) initial obduction; (2) exhumation/erosion; (3) deepening (4) shallowing, fault reactivation, and closure.

Comments on the text of the manuscript: Line 28: There are dozens of references that could be placed here. It is better to include "e.g." at the beginning of the reference list.

Done.

Line 28: You should give the definition for the Internal Hellenides as the term is not only geographical or spatial but also has a geodynamic meaning by dividing the Hellenides into two areas with different evolution during the alpine orogenesis. Also, the first letter

must be uppercase (Internal).

Corrected 'internal' to Internal - we also noticed this same mistake on line 35, which has also been corrected in the manuscript. Reviewer 1 also suggested we refer to the map or provide a definition of the Internal Hellenides, therefore we will make the positions of the Internal and External Hellenides more apparent in Fig.1 to address the concerns of both reviewers.

Line 34: What is the origin of this "Upper Cretaceous basin"? How was it created? Is it a single basin or more?

We will rephrase this part also considering the comment of reviewer 1.

Line 36: I think that the migration is towards the SW-SSW.

The migration direction depends on whether one is talking about present-day coordinates or not, therefore we will eliminate any confusion and simplify this sentence by replacing SSE with 'southward' in the text.

Line 41: What are these controversies? I believe it needs further analysis beyond a simple reference to "controversies" and the presentation of a figure (Figure 2). You need to clarify the problem that you want to solve with this work.

This is very similar to a point raised by reviewer 1 – we need clarify the controversies and the problem we wish to solve. The many and contrasting geodynamics models present in the literature source from the difficulties of connecting the Rhodope and the Pelagonian zone. This is a longstanding debate on the number and dimension of oceans in the Mesozoic Pindos-Vardar realm between researchers proposing a single unifying Early Jurassic Vardaric ocean that has been partly subducted, partly obducted and dismembered during successive tectonic events and researchers that embraced a multi-ocean early Jurassic scenario that led to several subduction zones. In these scenarios, the Cretaceous sediments were deposited on the eastern Pelagonian zone either within a Jurassic-Cretaceous passive margin or during a subsequent Cretaceous

tectonic event (compressional or extensional depending on the authors). These geo-dynamic interpretations are presented in Fig.2 and display the different positions of the Pelagonia-Vardar margin relative to the Alpine orogenic wedge after the Jurassic convergence. By studying the small Upper Cretaceous Kallipetra Basin that lies on the Pelagonia-Vardar 'suture zone', we can begin to address questions on if and how the Pelagonian-Vardar margin was deforming. Our study will ultimately provide constraints on the position of the eastern Pelagonian margin relative to the Alpine orogenic wedge, hence ruling out some of the geodynamic models so far proposed. Line 68: Add "e.g." at the beginning of the reference list as there are numerous works that could be cited here.

We will add 'e.g.'.

Lines 78-81: The area in which this stratigraphic gap has been described (Aptian-Albian) is very far from the study area and paleogeographically belongs to the wester nmargin of Pelagonian and not to the eastern. Furthermore, other researchers (e.g. Sharp and Robertson 2006) argue that the onset of sedimentation occurs during the Aptian-Albian north of the study area.

We agree that the area to which we refer to is far from the study area. Therefore, we will investigate descriptions of the Aptian-Albian hiatus and/or deposition from other studies such as Sharp and Robertson (2006) that are closer to our study area and edit the text accordingly.

Line 82: There are papers that describe older in age transgressive sediments which unconformably overlay the Pelagonian and Vardar units (e.g. Mercier 1968; Brown and Robertson 2004; Sharp and Robertson 2006; etc). See also my comment b.

We agree, but here we are referring to transgressive sediments to the south and not to the north. We will also rephase this part.

Line 83: You need to add more references here. There are numerous works to be cited

here, with primary data except from the synthetic work of Papanikolaou (2009).

Ok, we will add other works.

Line 92: Add "e.g." at the beginning of the reference list as there are numerous works that could be cited here.

We have added e.g. at the beginning of the reference list.

Line 95: Leucogneisses?

We will correct 'leucogneiss' to 'leucogneisses'.

Lines 95-96: Are you referring exclusively to the area west of the Kallipetra Basin or to the Pelagonian in general? If the latter is true you should add more references, as it is not only Schenker (2013) who describes the above lithologies. You could add "Schenker 2013 and references therein".

In this case, we are referring to and describing only the lithologies in the study area - hence just the area west of the Kallipetra Basin studied in Schenker (2013).

Line 111-112: The sentence 'the sediments belong.............as the Kallipetra bas-in ÌĞz causes confusion (see also previous comment c). What is called as Kallipetra basin? Is it the paleogeographic domain where the large thick Upper Cretaceous sediments were deposited or only the small basin under study?

The Kallipetra Basin is the small basin under study but could be correlated with other Late Cretaceous sediments found in nearby areas along the suture zone.

Lines 115-118: Please enter references as you seem to be referring to older works.

The work referenced here is Schenker et al 2015, we will add this to the manuscript

Line 141: What do you mean by the term "metapelitic zones"?

We mean 'diagenetic zones' or very low- to low-grade metamorphic zones. The term 'metapelitic zones' will be replaced by 'diagenetic zones'.

Lines 235-236: You argue that the fossils are deformed and reworked and are supplied by the VOC based only on the work of Schenker (2013). Apart from this study, I do not remember any other study that reports Lower Cretaceous sediments in the VOC. On the contrary, there are papers that support the start of deposition in Aptian-Albian(see also previous comment b). Even in your own work it is described that sediments of Kallipetra Formation with VOC form duplexes, so how are you convinced that the fossils belong to VOC and not to the Kallipetra formation? Îd'here are also studies that describe Upper Jurassic-Lower Cretaceous sediments unconformably on the VOC, which seal the tectonic emplacement of the VOC onto the passive margin of the Pelagonian. If you include those Upper Jurassic-Lower Cretaceous sediments in what you have named as Vardar Oceanic Complexes then you need to clarify that.

Thank you for your comment. We need to reassess the origin of these deformed/reworked fossils. The deformation and lack of preservation, along with their age distribution, still suggests that they are reworked and do not represent the depositional age of the unit from which they are found in. However, it is a possibility that they have been supplied from the very base of the Kallipetra Basin units that form the duplexes. We will reassess and adjust the manuscript accordingly to clarify the origin of the fossils.

Lines 311 and 312: Please correct the references. There is no Schenker (2014) in your reference list.

Schenker (2014) should be corrected to Schenker et al., (2014).

Line 415: Please enter reference as you seem to be referring to older work.

This work should be Schenker et al., (2015).

Line 449: The word "dramatic" has been struck through. I believe you need to delete that word.

Yes, dramatic needs to be deleted.

Lines 488-489: See my comment d. As in the following lines (490-492) you suggest a localized inversion that predated the start of the general convergence, you have to enforce your interpretation.

We will reinforce our interpretation with a series of sketches, as mentioned in our response to commend (d).

Lines 494-498: I suggest to delete this interpretation as you have already weakened it in the second sentence.

See reply on the heat source to reviewer 1.

Comments on the Figures 1. Figure 2 shows various models of evolution of the Hellenides in the Cretaceous, which are not analyzed in the manuscript and in the end there isn't any suggestion about them. Therefore, it does not offer anything substantial to this work and could be removed.

We will keep figure 2 but make sure we explain our study goals more clearly and refer back to the figure once we interpret our data, and we will make sure to be more specific on the controversies we would like to address (see reply on the scientific question to review 1).

2. In the geological map of figure 3 some things are not visible and difficult to distinguish, e.g. difficult to distinguish black dots from dark blue ones. Therefore, some symbols need to be magnified.

We, and Reviewer 1, agree that the dots are very small. We will make the dots much larger along with the text labels to make it more visible.

3. In the geological sections of Figure 8, there is a large number of faults. According to the manuscript and the map of figure 3, these are normal, thrust and strike- slip faults. In order for the reader to find out which is which, he must constantly resort to the map. Therefore, I suggest the relative slip of the fault-blocks should be plotted along the faults.

We appreciate having this brought to our attention. The relative slip of the fault blocks was plotted along the faults, however the reduction in size of the figure was not taken into account so the labels were no longer visible. We will make the relative slip symbols much larger and visible to the reader.

4. In figure 12 there is no legend explaining the symbols used to describe the different geological formations of the sketch. The sketch also gives a false impression that the basin has developed mainly east and northeast of the VOC. Perhaps the sketch should also include the western margin of the basin in order for the reader to have a complete picture. See also previous comment for 3D sketches.

We will create a series of sketches that show the evolution of the basin to replace the single snapshot displayed in Figure 12 and will be sure to add a legend. We will 'snapshot' the following phases of basin development - (1) initial obduction; (2) exhumation/erosion; (3) deepening (4) shallowing, fault reactivation, and closure.

Please also note the supplement to this comment:
https://se.copernicus.org/preprints/se-2020-106/se-2020-106-AC2-supplement.pdf

―――――――――――――――――――

---

## Author Response (AR1)

Dear Jonas Kley & others,

I am pleased to submit the revised version of our original research article entitled 'Birth and closure of the Kallipetra Basin: Late Cretaceous reworking of the Jurassic Pelagonian – Axios-Vardar contact (Northern Greece)' on behalf of all the authors.

We are very grateful for the positive responses and the constructive comments provided by both anonymous referees. Their suggestions have significantly improved our manuscript, and their confusion over some of the main points allowed us to re-write some major sections, edit appropriate figures, and create a new figure in order to more adequately explain our complex geologic interpretations. We believe that our revised manuscript fully addresses the concerns of both referees and hope you will consider it for publication in Solid Earth and the special issue 'Inversion tectonics – 30 years later'.

In this document are point-by-point responses to all referee comments from both referees, a list of relevant (major) changes to the manuscript and figures, and a marked-up version of the revised manuscript.

**Response to Interactive Comment by Anonymous Referee #1**

We are grateful for the positive response and constructive comments provided by the anonymous referee. They raise 3 main points: (1) the focus of the study needs to be addressed more clearly in the introduction and in the discussion/conclusions; (2) our discussion and reasoning on the tilted thrust fault should be developed further; (3) the extensive discussion on the origin of fluids in the fault zone is not the topic of this study. These are addressed below, and specific comments to individual points of the manuscript are provided. In the following, the referee comments are in italics, and we respond in regular font.

*Anonymous Referee #1*
*Dear Editor, I have read with great interest the work of Bailey and coauthors regarding the Kallipetra Basin in N. Greece. The manuscript is well written, and it presents new data and interpretations in connection to the geodynamics and the problems of ophiolite obduction in the Hellenides. The writing is clear and easy to follow. The authors have put a great effort to document the data and the field evidence related to this study and I have to admit that it is rare to see papers with such a level of detail when it comes to the primary data. However, I have a few comments related to the overall presentation and some of the conclusions of the study. The most important points that I can mention here are:*
*1) The focus of the study with respect to the general problematics of tectonic scenarios in the Hellenides must be addressed more clearly. This is because the study may look a bit "too regional" from the perspective of a researcher who is not familiar with Eastern Mediterranean geology.*

We have altered the introduction of our manuscript to more clearly address the focus of our study with respect to the 'controversies' of the Hellenides. Additionally, we have edited Fig.2 accordingly to address the problematics of the Hellenides tectonic scenarios. Please see our more detailed response on this issue below, in the response to a comment on l.41 by the referee.

*2) The authors present their view of the main contact being a tilted thrust towards the NNE. This is a very important point given the current discussion in the literature related to the ophiolite*

*obduction problem (Pindos vs Vardar etc). I would thus suggest that the authors develop the discussion and explain their reasoning a bit in more detail.*
We thank the reviewer for this suggestion. We have added a new figure that provides a clear presentation of our tilted thrust zone, and provided a more detailed reasoning in the manuscript in Section 5.6.

*3) The authors get into an extensive discussion about the importance of fluids vs viscous heating while at the same time they also admit that the evidence is not so clear. Since the discrimination of the additional source of heat which is required is not the topic of this study (and there has been no effort inquantify the arguments), I would suggest that the authors mention the possibilities andnot go in a specific discussion on the importance of a particular mechanism.*

Our data document an inverted metamorphism below a shear zone. The heating that produced this metamorphism occurred during deformation and reset the FT ages, permitting us to date deformation. We acknowledge that physical models are needed to reproduce and quantify length and timing of the observed metamorphism. Therefore, as asked by the reviewer, we have significantly reduced the extensive discussion in Section 5.5 so the reader can simply recognize the presence of heat along the thrust zone without getting deterred from the main conclusions of our study.

*The Specific comments follow below:*
*l. 27: Please add "e.g." in the reference list. There are numerous works to be cited here.*
Done.

*l. 28: Please define what is meant by "Internal" Hellenides, either by definition or by citation to the map.*
We have referred to the map and have edited Fig. 1 by labelling the Internal Hellenides and External Hellenides more clearly.

*l. 34: Please be more specific about the "Cretaceous Basin". Does it have a name? Is there in a particular location that you refer to?*
Here we refer to the Kallipetra Basin.
The Cretaceous basins that formed at the eastern Pelagonian margin and over the Axios/Vardar zone were first mapped at the large scale by Kossmat 1924. Many workers have found sparse Cretaceous sediments since then (e.g. Mercier and Vergely 1994, Sharp and Robertson 2006). Schenker et al (2015) brought clear evidence that the gneissic detritus in one of these basins (named in this contribution Kallipetra basin) is of Pelagonian and not of Rhodopian origin (e.g. Ricou and Godfriaux 1995).
We have rephrased this part of the text to: "...by the deposition of metamorphic Pelagonian detritus in a Late Cretaceous basin (Schenker et al. 2015) subsequently referred to as the Kallipetra Basin in this study."

*l. 37: I would suggest that "pulse" is not the right word here. It is known that the extension and basin formation in the Hellenides is diachronous and migrating southwards (see also Papanikolaou & Royden, 2007) for more details and the relevant literature.*
We have rephrased to: "Finally, from the Oligocene-Miocene the western Pelagonia was dissected by diachronous normal faults (Schermer et al., 1990; Lacassin et al., 2007; Coutand et al., 2014; Schenker et al., 2014) within a southward extensional deformation front that affected most of the Hellenides (e.g. Papanikolaou & Royden, 2007)."

*l. 41 (MAIN POINT): It is not clear what are the main features that you would like to address in all these contrasting interpretations. In terms of the sketches that are presented in Fig. 2 the focus of this work can be i) the position of Pelagonia, ii) the number of subduction zones etc. Therefore, I suggest you develop on the specific features that you want to address in more detail. In other words, please identify the problem/hypothesis and then explain why you chose to focus on this area to solve it/test it.*

The many and contrasting geodynamics models present in the literature source from the difficulties of connecting the Rhodope and the Pelagonian zone. This is a longstanding debate on the number and dimension of oceans in the Mesozoic Pindos-Vardar realm between researchers proposing a single unifying Early Jurassic Vardaric ocean that has been partly subducted, partly obducted and dismembered during successive tectonic events and researchers that embraced a multi-ocean early Jurassic scenario that led to several subduction zones. In these scenarios, the Cretaceous sediments were deposited on the eastern Pelagonian zone either within a Jurassic-Cretaceous passive margin or during a subsequent Cretaceous tectonic event (compressional or extensional depending on the authors). These geodynamic interpretations are presented in Fig.2 and display the different positions of the Pelagonia-Vardar margin relative to the Alpine orogenic wedge after the Jurassic convergence. We have subsequently edited Fig.2 to show the position of the Kallipetra Basin in the different geodynamic scenarios.

By studying the small Upper Cretaceous Kallipetra Basin that lies on the Pelagonia-Vardar 'suture zone', we can begin to address questions on if and how the Pelagonian-Vardar margin was deforming. Our study will ultimately provide constraints on the position of the eastern Pelagonian margin relative to the Alpine orogenic wedge, hence ruling out some of the geodynamic models so far proposed.

Thanks to the comments of both reviewers, we have adjusted the manuscript so as to better identify the problem we want to address, and to show the importance of the birth and the closure of the Kallipetra basin in the context of the Hellenides. This can be seen in our revised Introduction, discussion, and conclusion.

*l. 68: Please add e.g. in the citation list since the development of these basins were known already from the time of Brunn and Aubouin (1950-60s)*
Done.

*l. 77: "metamorphic ages of migmatites" should change to "zircon ages from the leucosomes from the migmatites".*
OK, we agree: the term proposed is more descriptive.

*l. 80: "of the wedge" Please rephrase so that you can be more specific on the kind of the wedge (e.g. accretionary, orogenic etc).*
We mean orogenic wedge and have rephrased accordingly.

*l. 93: Please add Brun & Sokoutis as well as Dinter & Royden for the Rhodope corecomplexes.*
This has been done.

*l. 95: leucogneiss -> leucogneisses*
Changed.

*l. 105: Please avoid terms that refer to processes which you cannot show (i.e. "hydraulically").*
The unit "hydraulically brecciated serpentinite" has been re-named to "Dark massive fractured to brecciated serpentinites" throughout the text and figures.

*l. 117: Please add reference to show who did this interpretation (after "basin").*
OK, it is the interpretation of Schenker et al 2015.

*l. 118: As above, please add reference at the end.*
Schenker et al 2015

*l. 127: "package" -> "pile"?*
We have replaced "sedimentary package" with "sedimentary sequence".

*l. 131: Please be specific because there are also other kinds of grade (i.e. ore grade). I suggest rewording as: "to determine grade..." -> to determine the metamorphic grade in low-grade metapelitic*
We agree with this suggestion, and the phrase "to determine grade" has been replaced with "to determine diagenetic grade".

*l. 141: What exactly do you mean by the "determination of metapelitic zones". I think you refer to the "metamorphic" zones. Right?*
Yes, we refer to low-grade metamorphic zones, so we have replaced 'metapelitic zones' with 'low grade metamorphic zones' in the manuscript.

*l. 168: As before, please remote the word "hydraulically".*
We have removed the term 'hydraulically' and changed it to '"Dark massive fractured to brecciated serpentinites", as mentioned in an above comment.

*l. 449: "dramatic" has been struck through. Please check the sentence.*
Thank you for alerting us to this, the word dramatic has been removed.

*l. 457: "and on viscosity". I would remove the specific mention to "and on viscosity"since any irreversible deformation mechanism would also contribute to shear heating (e.g. rate-independent plasticity)*
We agree with the reviewer and deleted "and on viscosity".

*l. 460: "With a <2cm/a the heat is..." This statement assumes that the movement is steady. Since this hypothesis cannot be supported by the present data, I would suggest removing this sentence.*
See comment below the following point.

*l. 465-467 (MAIN POINT): As before, the discussion around viscosity only, neglects the frictional part of the heat. Therefore, since this is not the main topic of this paper and there is no detailed analysis in this direction, I would remove specific conclusions related to the most-likely source and the magnitude of shear heating.*

As the reviewer has helpfully pointed out, the discussion around the specific magnitudes of heating related either to shear heating or advected hot fluids is highly hypothetical, and we therefore do not have adequate evidence to support one of the two sources of heat. Rather, the goal of this particular paragraph was to draw attention to the unusual inverse geothermal gradient and explore possibilities of how/why this formed.
Therefore, we have re-written, simplified, and shortened Section 5.5 'The inverted geothermal gradient in the Kallipetra Basin' to address the concerns of Referee #1.

*l. 470-471: As before, there is no evidence to suggest what is considered "normal" by the authors since: (i) the rheology does not have to be purely viscous, (ii) the motion does not need to be steady. Therefore, the suggestion of a particular heating mechanism is beyond the scope and the data presented in this study.*

We acknowledge that we have no evidence or data that addresses the convergence rates, viscosities, or plate velocities and therefore agree that the suggestion of particular heating mechanisms goes beyond the scope and data presented in this study. Therefore, we have re-written and shortened this section so that we only relate the observed inverse geothermal gradient to the closure (and timing of closure) of the Kallipetra Basin so that it remains in the scope of our study.

*l. 475: Why the direction of tectonic transport is related to the fluid flow. Assuming a fault zone as a region of high permeability is well established. However, I cannot see how the transport is related for this conclusion.*

We agree that transport is not necessarily related to this conclusion, therefore we have replaced "The overriding unit over the Kallipetra basin would have allowed fluid focusing and differential loading that caused any fluids to flow in the direction of tectonic transport" with "Differential loading from the overriding unit over the Kallipetra Basin could have focused fluids along the fault zone".

*l. 488-490: How did you conclude that this must be thrusting (MAIN POINT). Why not normal fault with top NE kinematics. Please explain your reasoning in more detail.*

The conclusion for thrusting to the NE came from the stratigraphic evidence, predominantly from the character of the rudist mounds (e.g. mound asymmetry, younging direction, and ophiolitic detritus) along with kinematic indicators. We see stacking of serpentinitic breccias on south-western flanks of rudist mounds, sourced from ophiolitic debris up slope. The absence of ophiolitic detritus on the northeastern mound flanks document a 'shadow' effect of the mounds with respect to a south-southwestern provenance of serpentinite clasts. The highest, and therefore youngest, mound is located at Asomata Quarry which is the most northeastern mound suggesting movement of the ophiolite from SSW to NNE. Part of the reason the rudist mounds are so interesting is that they tell us something about the tectonic activity the basin is experiencing without needing to observe the fault itself.

We understand that this is complicated and was difficult to grasp in the way we originally wrote the manuscript. In order to provide some clarification and to expand our reasoning in more detail, we have added a new figure that documents the opening and closure of the Kallipetra Basin, and the figure also compares features we would expect to see for both normal faulting and thrust faulting scenarios (New Fig. 12). We have also expanded our reasoning in Section 5.6.

*l.494: As before, since the authors already state in line 478 that the sources of heat are not clearly established. I would leave the interpretations out of this.*
See comments above pertaining to this issue. We have removed the interpretations of heat sources from this sentence.

*l. 494-496: These places are quite far from each other.*
Indeed they are. We have removed this sentence.

*l. 496-498: From Turonian to Campanian is more than 10 Myr. For a crust ~10km thick and standard thermal parameters, the conductive thermal relaxation timescale is ca1Myr. Therefore,*

*I do not think that the advective heat was maintained long enough to cause the heating. Therefore, I would suggest that the authors revise this sentence to defend or reject this conclusion.*
We have deleted this sentence as it directly follows from the previous sentence which was deleted in response to the reviewer comment above.

*Legend Fig. 3: "Dark blue circles", the samples are very small. Please use larger and more discrete symbols.*
We agree, very small - we have adjusted the figure so the sample location circles are much larger and a brighter color.

**Response to Interactive Comment by Anonymous Referee #2**

We thank the reviewer for their detailed and thorough review of the manuscript, which will allow us to significantly improve our manuscript. The referee's main point was addressing the lack of large-scale implications and comparison with neighboring areas with Upper Cretaceous sediments. We have addressed their main concerns and respond to their individual comments below. We believe that our revised manuscript and newly created or edited figures have now included further comparison to nearby regions and a clearer statement of our study goals and the controversies we wish to elucidate. In the following, the referee comments are in italics, and we respond in regular font.

*Anonymous Referee #2*
*This work deals with the paleogeographic and tectonic evolution during the Upper Cretaceous of an area of Continental Greece that belongs to the so-called Internal Hellenides. Little is known about the Cretaceous evolution of this sector of the Hellenides and many questions await answers. Apart from the number of oceanic basins, the polarity of the subduction zone, etc, there are questions about the origin, age, deposition paleoenvironment and the geodynamic significance of the Cretaceous sediments deposited unconformably on top of the obducted Vardar ophiolite complexes and the Pelagonian passive margin. Thus, this manuscript fills a significant gap in our knowledge of these issues.*
*It is a well-written and well-structured manuscript with a wealth of data clearly presented, but in the end, it leaves the reader partially dissatisfied. And this has to do mainly with the large-scale implications of the results and their comparison with other neighboring areas of the Internal Hellenides where Upper Cretaceous sediments are also observed. As the authors report, in order to elucidate part of the controversies, they studied this small Upper Cretaceous basin, but the part of their manuscript that refers to those is poorly developed. I believe that a better analysis of this would strengthen their work even more.*

*Based on that I have noted the following:*
  a) *The work that first described the Cretaceous sediments east of the Pelagonian (Almopias Zone) is not included in the reference list, although this work is about an area just north of the Kallipetra basin and gives detailed lithostratigraphic columns presenting their paleogeographic and tectonic evolution. This work is: Mercier, J., 1968. Etude geologique des zones Hellenides en Macedoine centrale(Grece). Ann. Geol. Pays Hell. 20 (792 pp.).*

We have entered in more detail the comparison of the Lower Cretaceous basin to the work of Mercier, Robertson, Ricou and others. We hope that by adding some detailed comparisons with

other Upper Cretaceous basins in nearby regions has strengthened the part of our manuscript that aims to elucidate the controversies.

b) *There is no comparison or correlation with other areas where the Cretaceous sediments are also observed. There could be a comparison apart from Mercier's work with the results of other papers, e.g. the paper of Sharp and Robertson (2006), who give anevolution model of a similar Upper Cretaceous basin north of the study area. Mercier(1968) places the beginning of the deposition of the Upper Cretaceous sediments in Aptian-Albian, while other researchers such as Sharp and Robertson and Galeos etal. (1994) describe even older aged sediments (Upper Jurassic). It could also be compared to other areas of the non-metamorphic Pelagonian, e.g. in Othrys Mt (Ferriere,1982) and Argolida (Baumgartner, 1985). It is important to comment on the age of onset of the deposition of the Upper Cretaceous sediments, as well as the age of the emplacement of ophiolitic complexes on them, highlighting the possible differences that may exist from region to region.*
*Ferriere J (1982) Paleogeographies et tectoniques superposees dans les HellenidesInternes au niveau de l'Othrys et du Pelion (Grèce). Soc Geol Nord Publ 8:1–970.*
*Galeos, A., Pomoni-Papaioannou, F., Tsaila-Monopolis, S., Turnsek, D. & Ioacim, C.1994. Upper Jurassic–Lower Cretaceous 'molassic-type' sedimentation in the westernpart of the Almopia subzone, Aridhea Loutra Unit (northern Greece). 7th Congress of the Geological Society of Greece, Thessaloniki, May 1994.*

We agree with this point (see reply above) and we have compared our units with others, specifically with ages of deposition and/or emplacement, located not too far North of our study area.

c) *The phrase "Upper Cretaceous basin" is used in two ways: either to describe the wider paleogeographic area where the Upper Cretaceous sediments were deposited or the small basin of Kallipetra. This dual use of the term confuses the reader. It must be made clear that the Kallipetra basin is part of a wider paleogeographic domain which, during the Upper Cretaceous, was the site of deposition of large thickness sediments.*

Thank you for bringing this to our attention - we have gone through the manuscript to make sure this is cleared up to eliminate any confusion. In line 34, for example, we refer to the Kallipetra Basin and have rephrased this part of the text: "...by the deposition of metamorphic Pelagonian detritus in a Late Cretaceous basin (Schenker et al. 2015) subsequently referred to as the Kallipetra Basin in this study", and have clarified other uses of 'Upper Cretaceous Basin' in our manuscript.

d) *An important key in the evolution of the basin is the origin of the fault that places the Vardar Oceanic Complexes (VOC) on the Upper Cretaceous sediments in the easternpart of the basin. According to the authors, the direction of tectonic transport of the VOC sealing Kallipetra Basin was from SSW to the NNE. It seems difficult that this transport can place the VOC on the sediments of the basin in a distance at least 4km into the basin and*

*westwards, as shown by the geological map in Figure 3 and the geological sections in Figure 8. This could happen if the VOC nappe crossed the entire basin from southwest to northeast. Also, in the map of figure 3 the fault is characterized as a reactivated thrust fault. This is not clearly described in the text except perhaps from the sentence in line 490. A much better analysis and documentation of the interpretation given is needed.*

We do not fully understand the argument in this comment, however this, along with a similar comment from Reviewer 1, alerted us to confusion over the reactivated thrust fault and direction of transport in our manuscript. We have made sure there is a better description and documentation of the reactivated thrust fault in section 5.6 'Sealing of the Kallipetra Basin and large-scale implications. We have also created a new figure with 2D sketches that show (1) the north eastward migration of the mounds is related to thrusting and not to normal faulting, (2) normal vs. inverted thrust, and (3) subsequent rotation of the fault into a 'normal' position.

e) *What is the origin of the basin and how is it associated with the growth of the Hellenides? Is it a fore-arc basin formed on top of an evolving accretionary wedge, is it a basin formed at the back of an orogenic wedge that collapsed due to underplating at its base, or is it a back-arc basin?*

Towards the end of the Kallipetra Basin timeline, the basin could be described as sediments accumulating in a foredeep generated ahead of an emplacing ophiolite. However, the basin formed under an extensional exhumation phase where there was a lot of erosion of both the Pelagonian continent and the Jurassic ophiolite, forming a depression. The upward deepening of the successions (before we shallow again due to the incoming ophiolite), suggests a phase of extension. In our area of focus, we see no evidence of the presence of a volcano so therefore the Kallipetra Basin was not a fore-arc or back-arc basin, and coeval volcanism is not known elsewhere. The basin formed on to of a suture zone. We have made sure our descriptions and discussion on both the opening and closure of the basin have been more clearly addressed in our revised manuscript, along with a new figure to alleviate some of the confusion. Furthermore, in section 5.6 of the revised manuscript, we discuss the location/position of the Kallipetra Basin with respect to the regional tectonics.

f) *The evolution of the basin could be captured by a series of sketches, which can be either NE-SW striking cross-sections or 3D sketches, beyond the snapshot of Figure12.*

We agree that this is a great idea and we have created a new figure, Fig. 12 in the revised manuscript, to help solidify some of our explanations and interpretations, especially with regards to your point (d). We have added a new figure that includes a snapshot of various times: (1) exhumation/erosion and opening of the basin; (2) deepening; (3) shallowing, mound growth, fault reactivation, and closure; (4) and tilting of the faulting contact and basin. Also in this figure is a comparison of top-to-the-NE **normal** faulting versus top-to-the-NE **thrusting** to show that we require top-to-the-NE thrusting to agree with our data and observations.

*Comments on the text of the manuscript:*
*Line 28: There are dozens of references that could be placed here. It is better to include "e.g." at the beginning of the reference list.*

Done.

*Line 28: You should give the definition for the Internal Hellenides as the term is not only geographical or spatial but also has a geodynamic meaning by dividing the Hellenides into two areas with different evolution during the alpine orogenesis. Also, the first letter must be uppercase (Internal).*

Corrected 'internal' to Internal - we also noticed this same mistake on line 35, which has also been corrected in the manuscript. Reviewer 1 also suggested we refer to the map or provide a definition of the Internal Hellenides, therefore we have made the positions of the Internal and External Hellenides more apparent in Fig.1 to address the concerns of both reviewers.

*Line 34: What is the origin of this "Upper Cretaceous basin"? How was it created? Is it a single basin or more?*

We have rephrased this part also considering the comment of reviewer 1.

*Line 36: I think that the migration is towards the SW-SSW.*

The migration direction depends on whether one is talking about present-day coordinates or not, therefore we will eliminate any confusion and simplify this sentence by replacing SSE with 'southward' in the text.

*Line 41: What are these controversies? I believe it needs further analysis beyond a simple reference to "controversies" and the presentation of a figure (Figure 2). You need to clarify the problem that you want to solve with this work.*

This is very similar to a point raised by reviewer 1 – we need clarify the controversies and the problem we wish to solve. The many and contrasting geodynamics models present in the literature source from the difficulties of connecting the Rhodope and the Pelagonian zone. This is a longstanding debate on the number and dimension of oceans in the Mesozoic Pindos-Vardar realm between researchers proposing a single unifying Early Jurassic Vardaric ocean that has been partly subducted, partly obducted and dismembered during successive tectonic events and researchers that embraced a multi-ocean early Jurassic scenario that led to several subduction zones. In these scenarios, the Cretaceous sediments were deposited on the eastern Pelagonian zone either within a Jurassic-Cretaceous passive margin or during a subsequent Cretaceous tectonic event (compressional or extensional depending on the authors). These geodynamic interpretations are presented in Fig.2 and display the different positions of the Pelagonia-Vardar margin relative to the Alpine orogenic wedge after the Jurassic convergence. By studying the small Upper Cretaceous Kallipetra Basin that lies on the Pelagonia-Vardar 'suture zone', we can begin to address questions on if and how the Pelagonian-Vardar margin was deforming. Our study will ultimately provide constraints on the position of the eastern Pelagonian margin relative to the Alpine orogenic wedge, hence ruling out some of the geodynamic models so far proposed.

We have edited Fig. 2 by outlining the position of the Kallipetra Basin with respect to the different geodynamic scenarios. We have changed our introduction to further elaborate on the controversies and our study goals, and referred back to the geodynamic interpretations in the Discussion. We expanded our discussion to include where our Basin is located compared to the regional tectonics.

*Line 68: Add "e.g." at the beginning of the reference list as there are numerous works that could be cited here.*

We have added 'e.g.'.

*Lines 78-81: The area in which this stratigraphic gap has been described (Aptian-Albian) is very far from the study area and paleogeographically belongs to the wester nmargin of Pelagonian and not to the eastern. Furthermore, other researchers (e.g. Sharp and Robertson 2006) argue that the onset of sedimentation occurs during the Aptian-Albian north of the study area.*

We agree that the area to which we refer to is far from the study area. Therefore, we have investigated descriptions of the Aptian-Albian hiatus and/or deposition from other studies such as Sharp and Robertson (2006) that are closer to our study area and edited the text accordingly.

*Line 82: There are papers that describe older in age transgressive sediments which unconformably overlay the Pelagonian and Vardar units (e.g. Mercier 1968; Brown and Robertson 2004; Sharp and Robertson 2006; etc). See also my comment b.*

We agree, but here we are referring to transgressive sediments to the south and not to the north. We will also rephase this part.

*Line 83: You need to add more references here. There are numerous works to be cited here, with primary data except from the synthetic work of Papanikolaou (2009).*

Ok, we have added more works: Mercier, 1968 and Mercier & Vergely, 2002.

*Line 92: Add "e.g." at the beginning of the reference list as there are numerous works that could be cited here.*

We have added e.g. at the beginning of the reference list.

*Line 95: Leucogneisses?*

We have corrected 'leucogneiss' to 'leucogneisses'.

*Lines 95-96: Are you referring exclusively to the area west of the Kallipetra Basin or to the Pelagonian in general? If the latter is true you should add more references, as it is not only Schenker (2013) who describes the above lithologies. You could add "Schenker 2013 and references therein".*

In this case, we are referring to and describing only the lithologies in the study area - hence just the area west of the Kallipetra Basin studied in Schenker (2013).

*Line 111-112: The sentence 'the sediments belong.............as the Kallipetra bas-in z causes confusion (see also previous comment c). What is called as Kallipetra basin? Is it the paleogeographic domain where the large thick Upper Cretaceous sediments were deposited or only the small basin under study?*

The Kallipetra Basin is the small basin under study but could be correlated with other Late Cretaceous sediments found in nearby areas along the suture zone.

*Lines 115-118: Please enter references as you seem to be referring to older works.*

The work referenced here is Schenker et al 2015, we have added this to the manuscript

*Line 141: What do you mean by the term "metapelitic zones"?*

We mean 'diagenetic zones' or very low- to low-grade metamorphic zones. The term 'metapelitic zones' has been replaced by 'diagenetic zones'.

*Lines 235-236: You argue that the fossils are deformed and reworked and are supplied by the VOC based only on the work of Schenker (2013). Apart from this study, I do not remember any other study that reports Lower Cretaceous sediments in the VOC. On the contrary, there are papers that support the start of deposition in Aptian-Albian(see also previous comment b). Even in your own work it is described that sediments of Kallipetra Formation with VOC form duplexes, so how are you convinced that the fossils belong to VOC and not to the Kallipetra formation? Îd'here are also studies that describe Upper Jurassic-Lower Cretaceous sediments unconformably on the VOC, which seal the tectonic emplacement of the VOC onto the passive margin of the Pelagonian. If you include those Upper Jurassic-Lower Cretaceous sediments in what you have named as Vardar Oceanic Complexes then you need to clarify that.*

Thank you for your comment. We have added the following paragraph to Section 4.2 to address the origin of these fossils:
"However, the fossils are deformed and not perfectly preserved, suggesting that they have been reworked and are supplied from elsewhere (Fig. S1). Indeed, Schenker (2013) discovered Lower Cretaceous Orbitolina in the VOC, located very close to the tectonic contact with the Kallipetra Basin. Late-Jurassic to Lower-Cretaceous limestones directly overlying the Pelagonian basement and the dismembered, eroded ophiolites are the probable source of these fossils (e.g. Brown & Robertson, 2004; Sharp & Robertson, 2006). Therefore, this sample is excluded from discussions about the depositional age of the Kallipetra Basin."

*Lines 311 and 312: Please correct the references. There is no Schenker (2014) in your reference list.*

Schenker (2014) has been corrected to Schenker et al., (2014).

*Line 415: Please enter reference as you seem to be referring to older work.*

This work should be Schenker et al., (2015).

*Line 449: The word "dramatic" has been struck through. I believe you need to delete that word.*

Yes, dramatic has now been deleted.

*Lines 488-489: See my comment d. As in the following lines (490-492) you suggest a localized inversion that predated the start of the general convergence, you have to enforce your interpretation.*

We have reinforced our interpretation with a series of sketches, as mentioned in our response to commend (d), and expanded on Section 5.6 that refers to the localized inversion.

*Lines 494-498: I suggest to delete this interpretation as you have already weakened it in the second sentence.*

See reply on the heat source to reviewer 1. The discussion around the specific magnitudes of heating related either to shear heating or advected hot fluids is highly hypothetical, and we therefore do not have adequate evidence to support one of the two sources of heat. Therefore we have deleted a significant portion of the Section 5.5 that used to address the heating mechanisms and sources of heat.

.

*Comments on the Figures*

1. *Figure 2 shows various models of evolution of the Hellenides in the Cretaceous, which are not analyzed in the manuscript and in the end there isn't any suggestion about them. Therefore, it does not offer anything substantial to this work and could be removed.*

We have kept figure 2 but made sure we explained our study goals more clearly in the revised manuscript, and we now refer back to the figure once we interpret our data in the discussion sction. We made sure to be more specific on the controversies we would like to address (see reply on the scientific question to review 1). We also added the hypothetical locations of the Kallipetra Basins on the various models of evolution of the Hellenides during the Cretaceous.

*2. In the geological map of figure 3 some things are not visible and difficult to distinguish, e.g. difficult to distinguish black dots from dark blue ones. Therefore, some symbols need to be magnified.*

We, and Reviewer 1, agree that the dots were very small. We have made the dots much larger and also changed their colors to make them more visible.

*3. In the geological sections of Figure 8, there is a large number of faults. According to the manuscript and the map of figure 3, these are normal, thrust and strike- slip faults. In order for the reader to find out which is which, he must constantly resort to the map. Therefore, I suggest the relative slip of the fault-blocks should be plotted along the faults.*

We appreciate having this brought to our attention. The relative slip of the fault blocks was plotted along the faults, however the reduction in size of the figure was not taken into account so the labels were no longer visible. We have made the relative slip symbols much larger and visible to the reader. We also changed the colors of the cross sections so they correspond correctly to the geologic map symbology.

*4. In figure 12 there is no legend explaining the symbols used to describe the different geological formations of the sketch. The sketch also gives a false impression that the basin has developed mainly east and northeast of the VOC. Perhaps the sketch should also include the western margin of the basin in order for the reader to have a complete picture. See also previous comment for 3D sketches.*

We have added a new figure that includes a series of sketches that show the evolution of the basin that includes the western margin of the basin. We also added colors to the 3D sketch (previous Figure 12, now Figure 13) that correspond to the colors used in the geologic map.

**List of relevant changes in the manuscript**

**Authors:** The author order was changed from:
Lydia R. Bailey, Vincenzo Picotti, Maria Giuditta Fellin, Filippo L. Schenker, Miriam Cobianchi, Thierry Adatte

to:
Lydia R. Bailey, Filippo L. Schenker, Maria Giuditta Fellin, Miriam Cobianchi, Thierry Adatte, Vincenzo Picotti

**Abstract**
- The abstract was re-written to de-emphasize the sources of heat responsible for the inverse geothermal gradient.

**1. Introduction**
- The introduction was re-written to better explain our study goals and hypothesis, and to clarify the controversies addressed in Fig. 2 and how they relate to our study.

**2.1 Large Scale tectonic setting**
- In addition to some minor alterations, we mention the studies of Sharp and Robertson (2006) and Mercier to elaborate on the large scale tectonic setting of our study area (lines 94 to 98).
- We add in more citations on lines 114-115.

**2.2 Main geologic features of the eastern Pelagonian margin**
- We changed 'hydraulically brecciated serpentinite' to 'dark massive fractured to brecciated serpentinites' in response to Referee 1, here and throughout the text.
- We add Schenker et al., 2015 to the citations when we refer to their work.

**3 Methods**
- Very minor changes in all sections.
- 'Diagenetic' added before 'grade' on line XX
- Metapelitic zones changed to 'low-grade metamorphic zones' on line XXX

**4.1 The Kallipetra Formation: facies and boundaries (in 4 Results)**
- Some additions were made to unit descriptions for clarification.

**4.2 Biostratigraphic data (in 4 Results)**
- A few sentences were added to explore alternative sources of reworked, deformed fossils that have been excluded from discussions about the depositional age of the Kallipetra Basin.

**5.1 Onset and evolution of the Kallipetra Basin**
- Schenker, 2014 was corrected to Schenker et al., 2014 (here, and throughout the text)
- We added comparisons to the Kallipetra Basin onset to nearby study areas to the north (Mercier, Sharp and Robertson, etc) and speculate further on the regional-scale setting and how that might have affected the onset of the Kallipetra Basin.

**5.2 The Rudist mounds: facies and evolution at the slope of the Kallipetra Basin**
- We directly compare our rudist bioherms to similar rudist mounds observed nearby, north of the study area, by Sharp and Robertson (2006).

**5.5 The inverted geothermal gradient in the Kallipetra Basin**
- We removed a significant portion of this section, as we do not have adequate data to speculate on the sources of heat that caused the inverse geothermal gradient, and this was not the main goal of our study.
- We added some clarifying sentences about the timing of the heating event.

**5.6 Sealing of the Kallipetra Basin and large-scale implications**
- This section saw the most significant changes and additions due to some confusion experienced by both referees. We made sure to clarify the reasoning and evidence behind the top-to-the NE tectonic transport *through thrusting* (to also accompany the new fig. 12) and compare our results and interpretations to nearby study areas, with speculations on why the Kallipetra Basin experienced a different tectonic history.
- We compare the inversion observed by the Kallipetra Basin to the regional-scale tectonics.
- We refer back to Fig.2, our introduction, and the controversies we aimed to clarify at the end of this section.

**Figures:**

- Any terms 'Vardar Ophiolitic Complex' in the figures were renamed to 'Vardar Oceanic Complex' to be consistent with the text.

- Figure 1 – The 'Vardar/Sava Zone' and 'Ophiolites' were combined into one unit labelled 'Ophiolites & associated sedimentary rocks'. We felt that there was confusion around the additional 'Ophiolite' unit, as the Vardar/Sava Zones are also ophiolitic units, so by combining them into one eliminates any of that confusion. We also added labels 'Internal Hellenides' and 'External Hellenides' to the map.
- Figure 2 – Different colors were given to the Pelagonia and Rhodope terrains to make their positions more clear to the reader. The position of the Kallipetra Basin was inserted on each geodynamic interpretation as a red bar.
- Figure 3 – The sample location circles (previously very small black and dark blue dots) were significantly enlarged and changed to red and yellow circles with black outlines to make them more visible. The SKB unit in the legend was renamed to 'Serp. & Kallipetra Fm. Breccia' in case the reader hadn't noticed yet what SKB stands for in the text. The cataclasites in the legend were renamed to more descriptive terms 'Pelagonian & serp. Cataclasite' and 'Serpentinite cataclasite'. 'Hydraulically brecciated serpentinite (& ophicalcite)' was renamed to 'Fractured to brecciated serpentinite (& ophicalcite)'. We changed the color of the Aliakmon River so it doesn't interfere with the mapped units. We made the colors of the units on the map brighter for clarity.
- Figure 4 – the resolution of the Sfikia Lower Road section (4a) was improved.
- Figure 7 – the resolution of the photographs were improved and we switched the positions of 7b and 7c to correspond to the order they are introduced in the text.
- Figure 8 – Some of the contacts and bed dips on the cross sections were corrected to more geologically realistic configurations that better match our observations. Arrows showing relative slip of fault blocks were enlarged. The mound, mound flank, and SKB facies were colored to correspond to the colors on the geologic map.

- Figure 11 – The figure was mirrored so the thrust was facing the opposite direction and the units were colored to correspond to the colors/legend used in the geologic map. Arrows that indicated movement of fluid along the fault zone were removed.
- Figure 12 (NEW) – We created a new figure that shows the onset and closure of the Kallipetra Basin and compares closure due to normal faulting to closure due to thrusting.
- Figure 13 (former Figure 12) – We colored the units in the figure to correspond to the legend used in the geologic map (Figure 3) and the rest of the figures to remain consistent.

[revised manuscript text omitted]